# Compression as Adaptation:
# Implicit Visual Representation with Diffusion Foundation Models

**Zongyu Guo** [* 1]   **Jiajun He** [* 2]   **Zhaoyang Jia** [1]   **Xiaoyi Zhang** [1]   **Jiahao Li** [1]   **Xiao Li** [1]   **Bin Li** [1]
**José Miguel Hernández-Lobato** [2]   **Yan Lu** [1]

 CODE   WEBSITE

## Abstract

Modern visual generative models acquire rich visual knowledge through large-scale training, yet existing visual representations (such as pixels, latents, or tokens) remain external to the model and cannot directly exploit this knowledge for compact storage or reuse. In this work, we introduce a new visual representation framework that encodes a signal as a function, which is parametrized by low-rank adaptations attached to a frozen visual generative model. Such implicit representations of visual signals, *e.g.*, an 81-frame video, can further be hashed into a single compact vector, achieving strong perceptual video compression at extremely low bitrates. Beyond basic compression, the functional nature of this representation enables inference-time scaling and control, allowing additional refinement on the compression performance. More broadly, as the implicit representations directly act as a function of the generation process, this suggests a unified framework bridging visual compression and generation.

## 1. Introduction

Large-scale visual generative models (Esser et al., 2024; Brooks et al., 2024) have demonstrated remarkable capabilities in synthesizing and manipulating visual content by learning rich visual knowledge from massive data. With the progression from image to video synthesis, these models are further expected to leverage such visual knowledge for higher-level visual reasoning (Wiedemer et al., 2025), moving toward unified and generalist vision foundation models.

Despite this rich visual knowledge embedded in visual gen-

erative models, visual signals themselves remain external to the model. Visual content is typically represented as pixels (Meng et al., 2022), latent variables (Rombach et al., 2022) or tokens (Esser et al., 2021), which must be separately encoded and then fed into the model to enable downstream interactions such as editing. This separation between internal model knowledge and external signal representations may lead to redundancy and inefficiency in representation, and limits the ability of the generative model to store or reuse past visual information over time.

To address this gap, we explore encoding visual signals as a function that describes the generation process, enabling synergy with large-scale generative models. This method—representing signals as a *function* rather than an explicit array—is commonly referred to as *implicit* (or *functional*) representations. An example is the implicit neural representations (INR). They model visual signals as continuous coordinate-based functions, typically parameterized by small multi-layer perceptrons (Mildenhall et al., 2021; Sitzmann et al., 2020). While effective in compressing individual signals (Dupont et al., 2021), they are trained largely from scratch and remain decoupled from the rich knowledge learned by large-scale models across diverse data.

In this work, we propose to compress visual signals as model adaptations to large-scale diffusion generative models via parameter-efficient fine-tuning (PEFT) techniques (Mangrulkar et al., 2022), such as low-rank adaptation (LoRA) (Hu et al., 2022). While LoRA has been widely used for tasks like personalized generation (Ruiz et al., 2023; Kumari et al., 2023), we show that it also encodes implicit representations when optimized for specific visual content. Leveraging the pretrained generative model as a visual knowledge prior, such implicit representations emphasize high-level visual semantics, making it particularly effective for compression. Moreover, we show that the induced adaptation parameters can be mapped into a single compact vector (Li et al., 2025), enabling compression of visual signals (such as an 81-frame video) into one vector.

We demonstrate that such implicit visual representations achieve strong perceptual video compression performance at extremely low bitrates on benchmarks, including the HEVC

---

[*]Equal contribution. Work done during internship at Microsoft. [1]Microsoft Research Asia [2]University of Cambridge. Contact to: Zongyu Guo <zongyuguo@microsoft.com>, Jiajun He <jh2383@cam.ac.uk>.

*Proceedings of the $43^{rd}$ International Conference on Machine Learning*, Seoul, South Korea. PMLR 306, 2026. Copyright 2026 by the author(s).

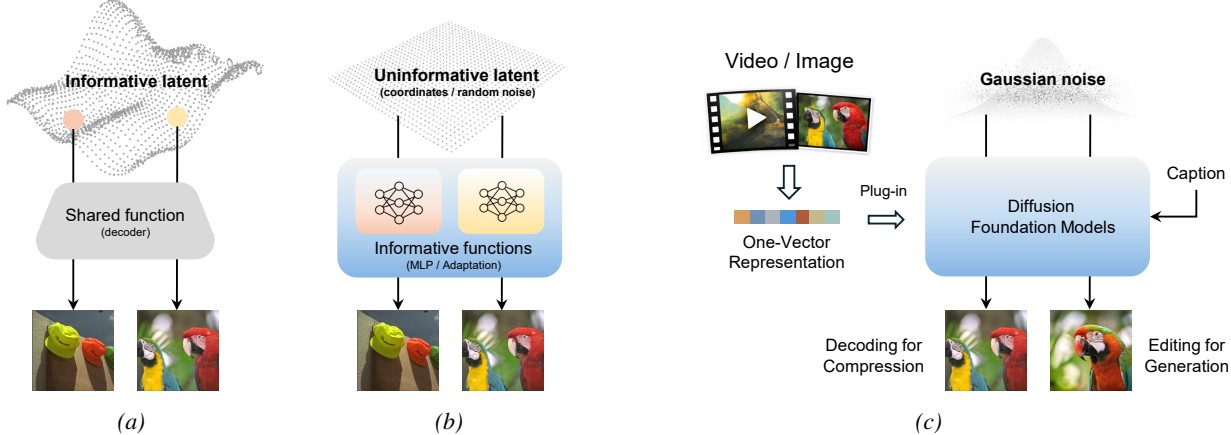

*Figure 1.* (a) Explicit representations by encoding signals into symbolic latent variables. (b) Implicit representations that encode signal information implicitly in functions. (c) One-vector adaptation of large-scale generative models can serve as implicit visual representations.

(Flynn et al.) and UVG (Mercat et al., 2020) datasets, achieving high visual quality even under extreme bitrate constraints. Representing visual content as a function embedded in a generative model provides additional flexibility that is unavailable to conventional codecs. In particular, the nature of functional representations enables inference-time control, allowing quality improvement through inference-time scaling. More broadly, the functional representations can be used beyond the reconstruction task and can serve as "visual memories" on specific identities, acting during the generation process, which suggests a step towards a unified framework to bridge compression and adaptive generation within pretrained generative foundation models.

In short, our core contributions include:

- We describe a framework that represents a visual signal as a function that generates the signal. Such implicit representations naturally leverage the rich knowledge embedded in large-scale visual generative models.

- We propose an approach that compresses the function into a single adaptation vector, achieving strong perceptual compression results on videos.

- We identify a key advantage of the functional representations: they can be flexibly controlled at inference time. Based on this insight, we introduce an inference-time scaling strategy for compression, which significantly improves reconstruction fidelity.

## 2. Background

### 2.1. Explicit and Implicit Representations

The distinction between *explicit* and *implicit* representations varies across research contexts. In some works such as implicit chain-of-thoughts (Deng et al., 2024; Zhu et al.,

2025), latent activations from neural networks are regarded as implicit representations, emphasizing their lack of direct interpretability. In contrast, much of the visual compression literature adopts a different perspective, where the distinction is defined by *how* signal information is represented: whether it is explicitly exposed as symbolic variables or implicitly encoded within a parametrized function (Dupont et al., 2021). In this paper, we follow the latter convention, where pixels, tokens, and latent variables are all treated as *explicit* representations.

**Explicit representations for visual compression**  Traditional lossy visual compression methods rely on explicit latent representations. A visual signal is encoded into a compact latent code, and reconstructed through a shared decoder. In this formulation, all sample-specific information is stored in the latent code, while the decoder remains fixed across signals. This paradigm underlies neural codecs (Ballé et al., 2017; Cheng et al., 2020; Li et al., 2021; Guo et al., 2021) based on variational autoencoders (VAEs) (Kingma & Welling, 2014), where compression performance is governed by the expressiveness and entropy of the latent space.

**Implicit representations as functions**  In this work, we use the term implicit representation to mean a functional encoding of a sample, i.e., representing a sample as a function. The function inputs could be coordinates (Stanley, 2007) or noise, which are shared or signal-agnostic, without any semantic information about the signal itself. Figure 1a and 1b illustrate such differences.

One specific example of the implicit representation is implicit neural representation (INR), which parametrizes the function with a small neural network. Such INRs enable compact and continuous representations of images (Dupont et al., 2021), videos (Chen et al., 2021), and 3D scenes (Mildenhall et al., 2021), which have motivated recent ex-

ploration of INRs for visual compression. However, existing INR-based compression methods (Guo et al., 2023; He et al., 2024a; Kwan et al., 2024) typically rely on small standalone networks, limiting their ability to recognize or exploit the rich visual knowledge, even in some recent works where INRs are trained with diffusion process (Gao et al., 2025b).

## 2.2. Diffusions and Flows

Our method builds on diffusion- and flow-based large-scale visual generative models (Wan et al., 2025; Wu et al., 2025). A diffusion model (Ho et al., 2020; Song et al., 2021) specifies a forward stochastic differential equation (SDE), gradually corrupting data to noise. We learn a *score* function by denoising score matching (DSM, Vincent, 2011) to reverse the noising process, mapping noise back to data.

Many recent vision models (Wan et al., 2025; Wu et al., 2025) are instead trained under a flow matching formulation (Lipman et al., 2023). The model learns a time-dependent vector field $v(x_t, t)$ and generates samples by solving an ordinary differential equation (ODE) defined by the vector field. With the common flow-matching formulation, the flow ODE admits an equivalent diffusion variant that preserves the same marginal densities, and the vector field can be directly related to the score. We detail the connection and the conversion between these formulations in Section C.3.

For the diffusion and flow-based vision foundation model, the generative procedure starts from random noise and then evolves the sample following the denoising SDE or the ODE. The samples they generated are determined by the entire dynamics, and hence, the generation process can be viewed as a representation of the generated sample.

# 3. Methods

This section presents our methods in details. We will begin with a high-level motivation and introduction to the framework of implicit visual representations.

## 3.1. Learning Implicit Visual Representations

**Motivations**   Our method starts from a simple observation: modern diffusion foundation models already contain a large amount of reusable visual knowledge in their weights, and they express this knowledge through a generation process. Strong priors about what "natural" images or videos look like are encoded in the pretrained weights and expressed through the entire generative procedure. Therefore, instead of compressing *what the visual signal is*, we compress *how to generate the visual signal*. In other words, rather than representing a visual sample explicitly (as pixels, latents, or tokens), we seek implicit visual representation as specified generation functions defined on top of the foundation model.

**Framework overview**   Given a visual signal $x$, we first use vision-language model (VLM) such as GPT-5.1 (OpenAI, 2025) to get a detailed caption $c$. Then, conditioned on this caption $c$, we learn model adaptation parameters on top of a frozen large-scale visual generative model. As the caption $c$ is fixed for each video, in the following presentation, we will drop it for simplicity. The resulting learned adaptation parameters therefore constitute an *implicit visual representation* of the input signal.

**Training objective**   We aim to fine-tune a diffusion (flow) model where the data distribution contains only $x$. Following the base models used in our experiments (Qwen and Wan; (Wu et al., 2025; Wan et al., 2025)) which adopt a vector-field parametrization, we learn a time-dependent vector field $v_\theta(x_t, t)$ with the flow-matching objective:

$$\mathcal{L}_{\text{FM}}(\theta) = \mathbb{E}_{t,\epsilon}\left[\left\|v_\theta(x_t, t) - (\epsilon - x)\right\|^2\right], \qquad (1)$$

where $\epsilon \sim \mathcal{N}(0, I)$, and $t \sim \text{Unif}[0, 1]$. Here, $x_t = (1 - t)x + t\epsilon$ denotes the linear interpolant between $x$ and $\epsilon$.

After training, at generation time, we start from a random sample $x_1 \sim \mathcal{N}(0, 1)$, and either integrate the corresponding flow ODE or simulate the associated diffusion SDE:

$$\mathrm{d}x_t = v_\theta(x_t, t)\mathrm{d}t \qquad (2)$$

$$\mathrm{d}x_t = \left(\frac{x_t}{1 - t} + 2v_\theta(x_t, t)\right)\mathrm{d}t + \sqrt{\frac{2t}{1 - t}}\mathrm{d}\overleftarrow{w}_t \qquad (3)$$

where $\mathrm{d}w_t$ represents the Wiener process (Brownian motion), and the arrow indicates the direction of the process. We derive the form of the SDE in Appendix C.3.

**Interpretation of the training objective**   The training objective in Equation (1) aims to find a function reproducing $x$. However, such a function is generally not unique: many different generation functions can all lead to the same final reconstruction. This raises a natural question: which function should we prefer? For compression, we seek the *simplest* function that achieves the desired reconstruction, in the sense of *deviating from the pretrained model as little as possible*. In fact, assuming the pretrained model and our model both follow the diffusion process (Equation (3)), we can motivate our objective from the minimum description length (MDL) perspective.

More concretely, a pretrained model induces a reference path measure $\mathbb{P}$ on SDE trajectories in the path space $C([0, 1], \mathcal{X})$. Our new model induces another path measure $\mathbb{P}'$ on the same path space. The expected excess "codelength" incurred when we encode a sample from $\mathbb{P}'$ using $\mathbb{P}$ as the coding distribution is the relative entropy $D_{\text{KL}}[\mathbb{P}'\|\mathbb{P}]$. Therefore, we want to find $\mathbb{P}'$ that deviates from the pretrained path measure $\mathbb{P}$ as little as possible while enforcing the terminal state $x$. Let $\{x_t\}_{t\in[0,1]} \sim \mathbb{P}'$ denote a trajectory

sampled from $\mathbb{P}'$, and $x_0$ for its state at $t \to 0$, we solve:

$$\min_{\mathbb{P}'} D_{\mathrm{KL}}[\mathbb{P}' \| \mathbb{P}] \quad \text{s.t.} \quad x_0 = x. \tag{4}$$

The optimal solution[1] is the base process $\mathbb{P}$ conditioned on the terminal event $x_0 = x$, which can be represented via a Doob's-$h$ transform of $\mathbb{P}$. Assuming the pretrained diffusion model is perfectly trained, the minimizer of Equation (1) recovers this solution. The proof follows directly from the definition of the Doob's-$h$ transform together with Bayes' rule; we include it in Appendix D.1.

**Benefit of functional representations** Functional representation brings two key benefits. First, it lets us directly leverage pretraining, as the compressed representation only needs to specify what is missing from the base model's default generation process. Second, a functional representation is not a fixed, one-shot code: the function itself naturally acts as a prior and remains controllable even after encoding. This means we can steer or refine generation while keeping the representation itself unchanged.

### 3.2. Compression with One-Vector Adaptations

While the above objective arises from a KL-constraint perspective on the path measure, modern neural networks are highly over-parametrized. Therefore, in practice, we still need to apply an explicit rate constraint over the adaptation.

**One-vector adaptation** First, instead of directly fine-tune the entire network, we only train an adaptation via Low-rank adaptations (LoRA) (Hu et al., 2022). For a pretrained weight matrix $\mathbf{W}_0 \in \mathbb{R}^{m \times n}$, LoRA models the weight update as a low-rank decomposition

$$\Delta \mathbf{W} = \mathbf{AB}, \tag{5}$$

where $\mathbf{A} \in \mathbb{R}^{m \times r}$ and $\mathbf{B} \in \mathbb{R}^{r \times n}$ with rank $r \ll \min(m, n)$. The new weight is given by $\mathbf{W} = \mathbf{W}_0 + \Delta \mathbf{W}$.

Although the rank $r$ is typically small, modern visual generative models contain a large number of layers. Equipping each with its own pair of LoRA matrices $(\mathbf{A}, \mathbf{B})$ keeps the total number of adaptation parameters substantial. To further reduce the number of parameters, we draw inspiration from the hashing trick used in neural network compression (Chen et al., 2015; Havasi et al., 2019; Müller et al., 2022): instead of storing separate adaptation parameters for each layer, we map all LoRA parameters into a single shared vector through a fixed projection, randomly generated by a pseudo-random number generator (PRNG). This design enforces parameter sharing across layers and encodes the entire adaptations into a compact vector $\mathbf{v} \in \mathbb{R}^{1 \times k}$. Note that

---

[1] Conditioning on $x_0 = x$ produces a measure $\mathbb{P}'$ that is singular w.r.t. $\mathbb{P}$ on $C([0,1], \mathbb{R}^d)$, so the KL on the full interval $[0,1]$ is infinite. We therefore interpret $D_{\mathrm{KL}}(\mathbb{P}' \| \mathbb{P})$ as the KL in restricted path space on $(0,1]$ (equivalently, on $[\tau, 1]$ for $\tau > 0$ with $\tau \downarrow 0$).

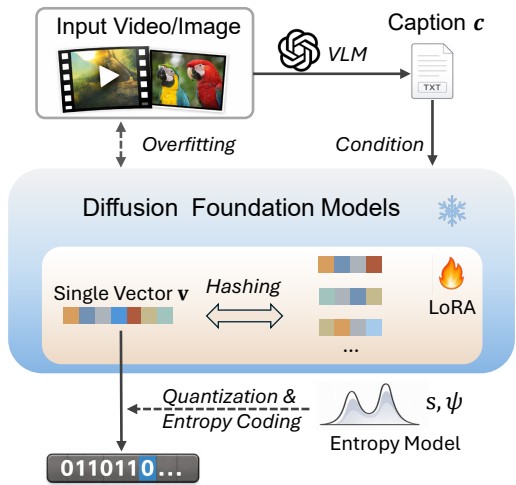

*Figure 2.* Framework for visual compression with a single vector that adapts the diffusion foundation model.

a closely related formulation has recently been proposed under the name Uni-LoRA (Li et al., 2025).

**Entropy constraint** While the one-vector formulation significantly reduces the number of adaptation parameters, the vector represented in floating-point precision (e.g., 32-bit or 16-bit) remains inefficient for extreme compression. We hence impose an explicit entropy constraint on the vector representation $\mathbf{v}$ and perform training-aware quantization.

Specifically, we (1) introduce a learnable scale parameter $s$ to normalize $\mathbf{v}$ before quantization, and (2) apply uniform quantization using rounding, with additive uniform noise during training to enable differentiable rate estimation, following Ballé et al. (2017). To model the bitrate, we adopt a factorized entropy model (Ballé et al., 2017) with learnable parameters $\psi$, which assumes independent distributions over vector elements.

With this entropy-constrained formulation, the one-vector adaptation can be quantized to approximately 1-3 bits per parameter while preserving good reconstruction quality. The additional parameters for the entropy model are transmitted in full precision: since the scale parameter is a scalar and entropy model relies on 8 distributional parameters (following Ballé et al. (2018)), the overhead is negligible.

### 3.3. Inference-Time Scaling and Control

As motivated in Section 3.1, representing a visual signal as a function provides an important benefit: after we obtain the one-vector adaptation, we can still control the generation process to enable modifications, improvements, or trade-offs, without changing the one-vector adaptations itself.

In this section, we highlight two representative directions that illustrate this benefit: (1) leveraging the idea

of inference-time scaling, we can obtain substantial reconstruction gains while incurring negligible overhead in code length; (2) beyond compression, the adaptations can be repurposed as an instance-level memory, enabling downstream reuse for personalized generation. Additionally, in Appendix B.1, we demonstrate that by intentionally terminating the generation process early, we can achieve an inference-time distortion–perception (DP) trade-off for free. We now describe these two points in more detail.

**Inference-time scaling with sampling** Inference-time scaling (Ma et al., 2025) for diffusion models uses extra computation at decode time to improve sample quality. A common strategy is to branch the SDE trajectory by drawing multiple candidates at each denoising step, then apply selection procedures such as sequential Monte Carlo (SMC) (Wu et al., 2023; Singhal et al., 2025; He et al., 2026) or search (Zhang et al., 2025) to retain the most promising candidates.

When we compress data via a functional representation, these inference-time scaling ideas carry over seamlessly: after the encoder obtains the LoRA adaptation, they can run the generation process using a pseudo-random number generator (PRNG) shared with the decoder. At every time step, the encoder samples multiple particles and then selects the most promising one. The selected particle is then encoded by its index using a small amount of side information. The decoder can then deterministically reproduce the same chosen particle at every step using the adaptation and the shared PRNG. Note that the scaling happens mostly on the encoding side, while decoding remains relatively fast; hence this can be viewed as *encoding-time scaling*.

Let us now make the algorithm more concrete to our setting. First, we now assume both the encoder and the decoder have access to the vector field $v_\theta$, parametrized by the pretrained generative model, plus our compressed adaptation. Then, in order to branch the denoising trajectory with multiple particles, we adopt the SDE formulation in Equation (3). We integrate the SDE numerically by Euler-Maruyama method. Concretely, we discretize the entire time horizon in $N$ steps, with boundaries $t_0 = 0 < t_1 < \cdots < t_N = 1$ and step size $\Delta t = |t_n - t_{n-1}|$. Each denoising step is given by a Gaussian transition kernel:

$$p(x_{t_{n-1}}|x_{t_n}) = \mathcal{N}(x_{t_{n-1}}|\mu_\theta(x_{t_n}), \frac{2t_n}{1 - t_n}\Delta t) \quad (6)$$

$$\mu_\theta(x_{t_n}) = x_{t_n} - \left(\frac{x_{t_n}}{1 - t_n} + 2v_\theta(x_{t_n}, t_n)\right)\Delta t \quad (7)$$

As this distribution is available to both the encoder and the decoder, the encoder can use the shared PRNG to draw multiple samples and select the best candidate. The remaining question is: what criteria should the encoder use to determine the best sample?

To answer this, let's recall the training objective in Equation (1). Its minimizer actually enjoys an analytical form:

$$v^*(x_t, t) = \epsilon - x \overset{x_t = (1-t)x + t\epsilon}{=\!=\!=\!=\!=\!=\!=\!=\!=\!=} (x_t - x)/t \quad (8)$$

For any input $x_t$ and $t$, the encoder can access this form, as it is then just a linear function of $x$. This means that the encoder actually access to the optimal denoising kernel:

$$p^*(x_{t_{n-1}}|x_{t_n}) = \mathcal{N}(x_{t_{n-1}}|\mu^*(x_{t_n}), \frac{2t_n}{1 - t_n}\Delta t) \quad (9)$$

$$\mu^*(x_{t_n}) = x_{t_n} - \left(\frac{x_{t_n}}{1 - t_n} + \frac{2(x_{t_n} - x)}{t_n}\right)\Delta t \quad (10)$$

Therefore, the encoder can perform *importance sampling*, with $p(x_{t_{n-1}}|x_{t_n})$ as the proposal and $p^*(x_{t_{n-1}}|x_{t_n})$ as the target, to select the best particle. Precisely, we draw $M$ samples $x_{t_{n-1}}^{(1:M)}$ from $p(x_{t_{n-1}}|x_{t_n})$, and then select an index from a categorical distribution over $1, \ldots, M$, where each probability is proportional to the importance weight

$$w^{(m)} \propto \frac{p^*(x_{t_{n-1}}^{(m)}|x_{t_n})}{p(x_{t_{n-1}}^{(m)}|x_{t_n})}. \quad (11)$$

This ensures the selected one follows $p^*(x_{t_{n-1}}|x_{t_n})$ when $M \to \infty$. The encoder repeats this sequentially for $n = N, N - 1, \ldots, 1$, until the final time step. We summarize the procedure in Algorithm 1. This algorithm resembles Sequential Monte Carlo (SMC, Del Moral et al., 2006), while only a single particle rather than the full particle population is retained at each step.

Notably, this procedure can also be viewed as a variant of the Diff-C algorithm (Theis et al., 2022), where the importance sampling with a fixed shared seed is a realization of the relative entropy coding (Havasi et al., 2019; Theis & Ahmed, 2022). Therefore, we can interpret our inference-time scaling algorithm from two complementary perspectives: (1) from the inference-time refinement perspective, we perform importance sampling to refine the learned generative dynamics induced by $v_\theta$; and (2) from the Diff-C perspective, we use the diffusion with adaptation as a stronger prior, thereby reducing the complexity of relative entropy coding. We include more background on relative entropy coding and discussion on the connections in Appendix C.4-C.6.

**Broader applicability: Functional representation via adaptation for generation** The functional representations in our framework are stored directly as parameter adaptations attached to a generative model, allowing visual information to persist for new generation. The adaptations are learned jointly with captions that serve as semantic anchors, linking specific visual entities to adaptations. Even when the captions are changed, the model still remembers the entities via the adaptations. In this sense, it is consistent with the LoRA-based customization (Ruiz et al., 2023; Kumari et al.,

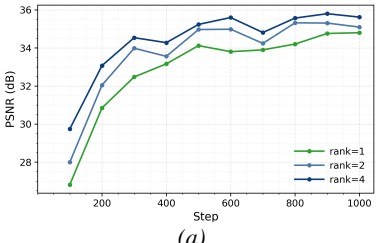 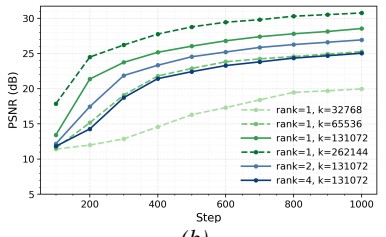 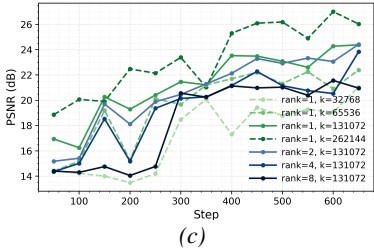

*(a)*        *(b)*        *(c)*

*Figure 3.* Reconstruction quality v.s. training step. (a) Common LoRA representations with different ranks for image *Kodim03* from Kodak dataset (Kodak, 1993). (b) One-vector representations for image *Kodim03*, varying LoRA rank and vector size after hashing. (c) One-vector representations for video *Beauty* from UVG dataset (Mercat et al., 2020).

2023; Ren et al., 2024; Meral et al., 2025), potentially enabling editing and composition via text prompts. It supports both faithful reconstruction and potentially flexible editing, bridging compression and generative modelling.

# 4. Experiments

## 4.1. Image and Video Representation

We first evaluate the representation capability of the proposed implicit visual representations. Following the standard LoRA setup (Hu et al., 2022), we fine-tune all the transformer blocks in the frozen generative model by adapting query, key, value, and output projection layers to fit a given visual signal. Both the LoRA rank and the number of fine-tuning iterations are adjusted for studying their impact on reconstruction quality. Detailed experimental settings are described in Appendix E.1.

As shown in Figure 3a, increasing the common LoRA rank consistently improves reconstruction quality, while even a minimal setting of rank = 1 is sufficient to achieve reasonable reconstructions. Second, reconstruction performance exhibits a clear upper bound, which we attribute to the inherent information loss introduced by the tokenizer of the underlying image or video generative model.

**One-Vector Representation** We further demonstrate that a single vector $\mathbf{v} \in \mathbb{R}^{1 \times k}$, obtained by hashing the optimized LoRA parameters, can effectively encode the visual content of either a single image or an 81-frame 480p video. As illustrated in Fig. 3b and Fig. 3c, increasing the vector dimensionality $k$ consistently improves reconstruction quality. Interestingly, when the vector size $k$ is fixed, we observe that increasing the LoRA rank leads to degraded reconstruction performance. This counter-intuitive behavior highlights a non-trivial interaction between LoRA capacity and the one-vector encoding process. We hypothesize that higher-rank LoRA adaptations introduce denser and more entangled parameter updates, which become difficult to preserve under a fixed-size hashing scheme.

## 4.2. Perceptual Video Compression

After representing the visual signal into a single-vector adaptation, we can further quantize this vector and apply entropy coding, yielding an extremely low-bitrate visual compression framework. Under this setting, the compressed representation prioritizes to boost perceptual reconstruction quality when decoded through a powerful generative model. We refer to this compression framework as *Vision (Video) in One Vector (VOV)*. Here, we demonstrate that VOV achieves strong perceptual video compression performance compared with existing video codecs.

**Settings** We compare **VOV** against representative neural video codecs, including both MSE-optimized (Jia et al., 2025) and perceptually optimized methods (Qi et al., 2025). For reference, we also report statistics from traditional video codecs, including H.265/HM (HM, 2021) and H.266/VTM (VTM, 2021). Benchmark datasets include the UVG (Mercat et al., 2020) and HEVC B/C/E datasets (Flynn et al.). We adopt Wan-2.1 (1.3B) (Wan et al., 2025) as the base video generative model. Due to the default generation constraints of the model, videos are processed at a resolution of $832 \times 480$ with 81 frames. For fair comparison, all benchmark videos are center-cropped and resized to 480p, and evaluated using 81-frame clips across all methods. Training details for compression are described in Appendix E.2.

**Metrics** We report DISTS (Ding et al., 2020), FVD (Unterthiner et al., 2018), Peak-Signal-to-Noise Ratio (PSNR) as primary evaluation metrics. We note that conventional pair-wise reference-based metrics such as PSNR are not well suited for the extremely low-bitrate regime considered here, as they fail to reflect perceptual quality when reconstructions are generated by stochastic generative processes. We also provide LPIPS (Zhang et al., 2018) results in Appendix F for completeness.

**Comparisons** As shown in Fig. 4, the proposed VOV achieves strong compression performance on the UVG dataset when evaluated using perceptual metrics such as DISTS and FVD. Moreover, VOV with inference-time scaling consistently yields substantial gains with only a marginal increase in bitrate, which suggests the scaling technique im-

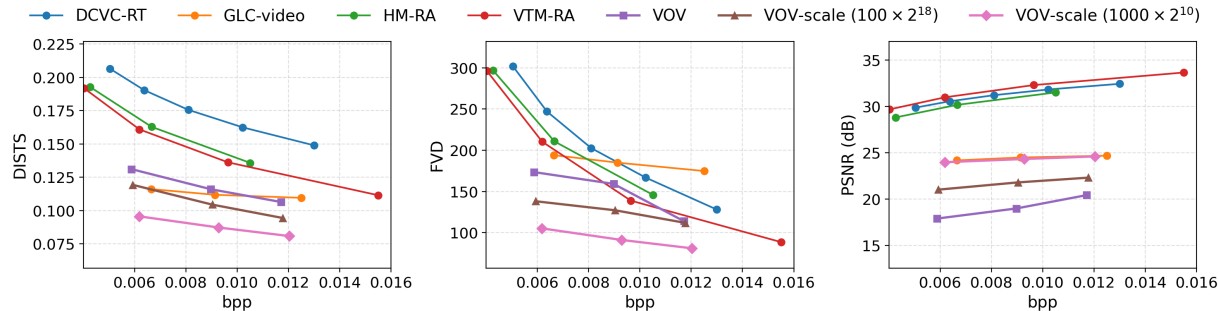

*Figure 4.* Comparisons with existing video codecs on UVG. For DISTS and FVD, lower is better. For PSNR, higher is better.

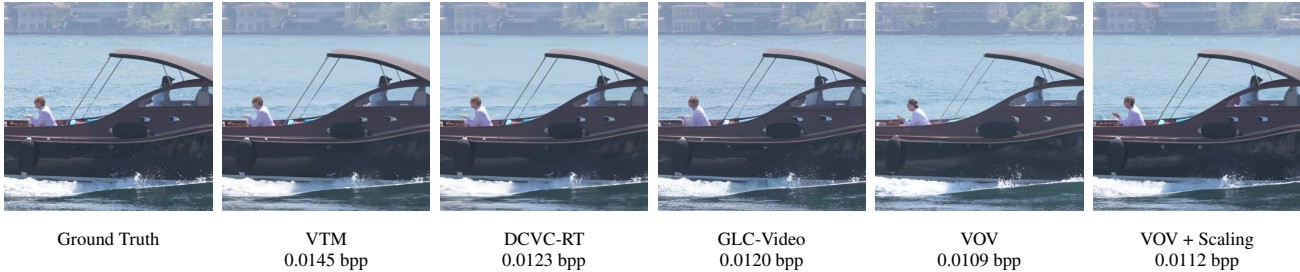

| Ground Truth | VTM 0.0145 bpp | DCVC-RT 0.0123 bpp | GLC-Video 0.0120 bpp | VOV 0.0109 bpp | VOV + Scaling 0.0112 bpp |

*Figure 5.* Visual comparisons with different methods. More results are in Appendix F.3 and supplementary materials.

proves the compression fidelity. Although the base VOV exhibits relatively low PSNR values, this does not contradict its strong perceptual performance. Pixel-wise fidelity metrics are often weakly correlated with human visual quality, which therefore are unreliable for extremely low bitrate video compression. This advantage is more clearly reflected in the qualitative comparisons shown in Fig. 5, where VOV reconstructs visually plausible structures and fine details that are missing in competing methods.

We further emphasize that temporal smoothness is a critical factor influencing perceptual video quality. By leveraging the temporal priors and rich visual knowledge embedded in video diffusion foundation models (Wan et al., 2025), videos decoded by VOV exhibit improved temporal consistency and reduced flickering artifacts, resulting in a more coherent viewing experience. More decoded frames and videos are provided in the Appendix F.3 and supplementary material. Note captions and entropy parameters are also compressed and transmitted. The bitrate of them are included in the curves, which are negligible ($< 1\%$).

### 4.3. Comparison with Implicit Neural Representations

We also compare the proposed method against the previous implicit neural representation-based codecs. As shown in Figure 10 in the Appendix, we report the results in terms of PSNR, DISTS and LPIPS by comparing with NVRC (Kwan et al., 2024) on UVG videos. We can observe a clear improvement from our method, particularly in terms of perceptual metrics (DISTS and LPIPS). The PSNR performance of our method is also competitive and even surpasses

NVRC at extremely low bitrates. This indicates that the diffusion backbone provides richer information.

### 4.4. Inference-Time Scaling

We now explore the inference-time scaling approach we proposed in Section 3.3. Note that we can scale along two axes: (1) we can branch the denoising process with multiple samples; and (2) we can increase the number of denoising steps. The first strategy only affects the encoding stage, while the second strategy increases the number of network evaluations and therefore incurs additional computational cost during both encoding and decoding.

In Figure 6, we investigate the performance with different scaling strategies on the YachtRide video from UVG dataset. Without branching (i.e., no SDE-based multi-sample selection per step), increasing the number of denoising steps yields almost no improvement. In contrast, introducing multiple samples per step substantially improves performance across all metrics, and using more samples leads to consistently better quality. One exception occurs at 50 steps, where using multiple samples performs worse. This is because the SDE sampler introduces larger discretization numerical error than the corresponding ODE sampler.

Additionally, when using multiple samples per step, increasing denoising steps also becomes effective — often more effective than further increasing the per-step sample size. In Figure 6, comparable gains can be obtained either by scaling the per-step candidates from $2^{10}$ to $2^{18}$, or by simply doubling the number of denoising steps. This gain, however,

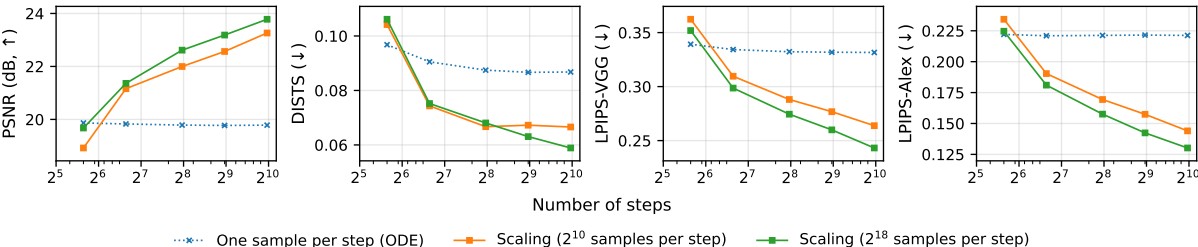

*Figure 6.* Reconstruction quality by inference-time scaling with different number of steps and different sample sizes per step.

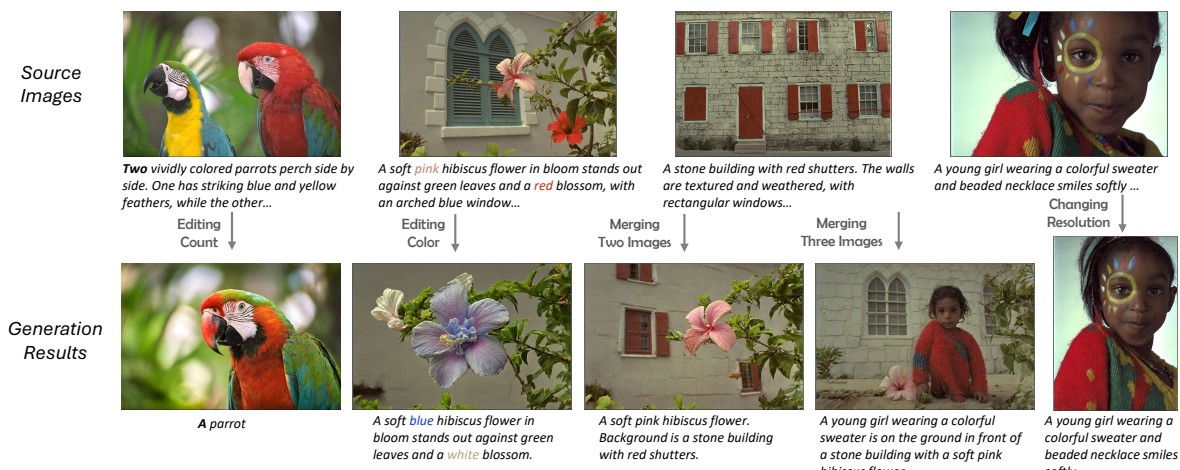

*Figure 7.* Use LoRA-based representations for personalized generation, such as image editing or merging. Base model is the text-to-image diffusion foundation model (Wu et al., 2025). Note that the original base model does not accept visual input.

comes at the cost of increased network evaluations. Therefore, although the performance trend has not yet saturated, which suggests that further increases in network evaluations could yield additional gains, we do not consider such scaling to be practical and in our main results we restrict the number of denoising steps to fewer than 1000.

In Figure 4, we compare our method against baselines under two inference-time scaling budgets. The first setting uses 100 denoising steps with $2^{18}$ samples per step, while the other setting uses 1000 denoising steps with $2^{10}$ samples per step. The first setting corresponds to only encoding-time scaling, while the second requires more compute in both encoding and decoding. Our approach with scaling becomes highly competitive with the baselines while still incurring an acceptable encoding and decoding cost, highlighting the advantage of functional representations.

**Pure scaling also achieves strong compression performance** We further consider a pure inference-time scaling strategy that directly optimizes the sample index using the original pretrained diffusion model, without relying on a one-vector adaptation. As shown in Figure 11, this approach can even achieve stronger compression performance, especially when a large scaling budget is used. However, this performance comes at the cost of many more denoising steps, making encoding and decoding substantially more

expensive. On the other hand, the learned adaptation makes the decoding process lightweight and can also achieve compact compression without scaling. Moreover, pure scaling only encodes a discrete and fixed sample index, whereas the one-vector representation provides a continuous and functional representation that better supports editing, control, and potentially amortization.

### 4.5. Distortion-Perception Trade-off via Early Stopping

The flexibility of functional representation goes beyond inference-time scaling. In Section B.1, we discuss that one can stop earlier in the denoising process and use Tweedie's formula to obtain the final reconstruction. We prove in Proposition B.1 that by varying the stopping time, the reconstruction error will first decrease and then increase. On the other hand, the perception metrics will monotonically improve, achieving an interesting distortion-perception trade-off. We explain and demonstrate this trade-off in detail in Sections B.1 and B.2.

### 4.6. Bridging Compression and Generation

As discussed at the end of Section 3.3, the proposed implicit visual representation directly modulates the generation function of a diffusion-based foundation model. After optimizing the LoRA-based adaptation for a given visual

signal, the semantic relationship between visual entities and textual prompts is effectively re-established. As a result, the adapted model not only can reconstruct the original signal, but also serves as visual memory that supports personalized editing or merging the encoded visual content.

To evaluate this capability, we conduct demonstrative editing experiments on a set of images from the Kodak dataset (Kodak, 1993). Detailed settings are in Appendix E.3. After training the LoRA-based (not one-vector representation) implicit representations and injecting them into the frozen diffusion foundation model, we modify the text prompt at inference time to perform personalized editing and visual composition. As shown in Figure 7, simple prompt changes enable meaningful semantic edits, such as editing colors or numbers, merging images and changing the resolution, while mostly preserving the coherence.

In Figures 12 and 13, we show that such editing is also possible with the one-vector representation and can extend to video as well. However, the effectiveness of this editing is limited by the model's capability and may reflect biases present in the training data. For instance, when modifying the model's hair color from blonde to black, the generated face may also shift to appear Asian.

**Generation with new prompts**   We also analyze the general generation ability when using the parameterized visual memory. We first study the generation with totally irrelevant prompts after overfitting. For the image in Figure 12, we can see that the general generation quality is largely preserved when the prompt is unrelated. However, for video, as we can see from Figure 14, the generation can be influenced by similar objects, visual styles, or motion patterns. This indicates that while the model is able to disentangle certain factors in the representation, it does not fully disentangle all attributes, and some remain correlated. Due to the different performance of images and video, we hypothesize that this limitation is largely due to the capacity of the base model.

In Section B.6, we also provide a preliminary analysis of generation with relevant prompts. We observe a trade-off between memorization and generation even for the image model: as overfitting (memorization) improves, the generation under relevant prompts becomes increasingly affected.

## 5. Conclusion

In this work, we propose a framework that represents a visual signal as a function describing its generation process, and compresses this function into a compact adaptation vector on top of a large-scale visual generative model. This framework leverages the knowledge learned by the pretrained generative model, enabling compact representation and efficient compression. We further identify and demon-

strate a key advantage of our approach—and, more broadly, functional representations: the function remains flexible at inference time, enabling further refinement and control.

**Limitations**   (1) Since the reconstruction is generated through a powerful diffusion prior, it is inherently limited by the capacity of the diffusion foundation model. **The decoded signal may hallucinate visually plausible details that are not faithful to the original input**, especially under extremely low-bitrate settings or for fine-grained structures such as text, faces, or small objects. (2) Similar to INR-based compression methods, **the encoding process (*i.e.*, overfitting the implicit representation) incurs an extremely long time**. Please see in Appendix F.1 for the detailed coding complexity. (3) **The correlations between adaptation parameters are captured through a randomly assigned hashing map**, which can be less effective. (4) Finally, the proposed method is sensitive to the architecture of the underlying diffusion. **Scaling to a larger backbone does not necessarily lead to better compression performance**. When increasing the model size, the LoRA adaptation size also needs to be increased, and then it may contain substantially more redundant or correlated parameters. As a result, the benefit of a stronger generative prior may be offset by a less efficient compressed adaptation representation. We did not analyze this scaling behavior in the current work. However, in our editing experiments, we found it necessary to increase the size of the one-vector representation for Wan-2.1-14B to obtain satisfactory adaptation quality. This partially suggests that the RD performance may degrade substantially when moving to much larger diffusion backbones with the current hashing map strategy.

**Future Directions**   A promising future direction is to learn an amortized encoder and decoder for the vector representation. This would substantially reduce the encoding cost by avoiding per-signal overfitting. Moreover, such learned encoders and decoders could better capture correlations among adaptation parameters than the current randomly assigned hashing map, largely improving the compression.

Second, future work should more systematically study and mitigate hallucination in generative compression. Since the reconstruction is produced through a diffusion prior, the model may introduce visually plausible but input-inaccurate details, especially at extremely low bitrates.

Third, it is important to analyze how the proposed framework scales with larger diffusion foundation models. As we discussed above, simply increasing the backbone size may not directly improve compression performance unless the compressed adaptation representation can efficiently capture the relevant parameter correlations. Understanding this behavior, especially with the learned encoder and decoder, remains an important direction for future work.

## Acknowledgment

JH acknowledges support from the University of Cambridge Harding Distinguished Postgraduate Scholars Programme. JMHL acknowledges funding from AI Hub in Generative Models, under grant EP/Y028805/1.

## Impact Statement

This paper presents work whose goal is to advance the field of machine learning. There are many potential societal consequences of our work, none of which we feel must be specifically highlighted here.

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

## Appendix Contents

# A. Algorithm

---

**Algorithm 1** Encoding-Time Scaling via Importance-Sampling

---

**Require:** Target $x$; adapted vector field $v_\theta$; time grids $0 = t_0 < \cdots < t_N = 1$; number of candidates $M$; shared PRNG.
**Ensure:** Indices $\{i_n\}_{n=1}^N$.
 1: Sample $x_{t_N} \sim \mathcal{N}(0, I)$ with shared PRNG.
 2: **for** $n = N, N-1, \ldots, 1$ **do**
 3:     Draw $M$ candidates $x_{t_{n-1}}^{(1:M)} \sim p_\theta(x_{t_{n-1}} \mid x_{t_n})$.
 4:     **for** $m = 1, \ldots, M$ **do**
 5:         Compute importance weight
$$w^{(m)} = \frac{p^*(x_{t_{n-1}}^{(m)} \mid x_{t_n})}{p_\theta(x_{t_{n-1}}^{(m)} \mid x_{t_n})}.$$
 6:     **end for**
 7:     Sample index $i_n \sim \text{Categorical}(w^{(1:M)})$.
 8:     Set $x_{t_{n-1}} \leftarrow x_{t_{n-1}}^{(i_n)}$.
 9:     Store $i_n$.
10: **end for**

---

# B. Additional Experiments and Findings

In this section, we demonstrate and investigate several aspects of our approaches. These findings illustrate interesting properties of our approach, and we hope to have a deeper understanding on our approach and inspire future works by describing these findings.

## B.1. Distortion-Perception Trade-off via Early Stopping

In this section, we discuss an interesting finding, showing that our approach has an inherent Distortion-Perception trade-off if stopping earlier during the generation process.

To understand this, let's recall the analytical form of $v^*$ in Equation (8). This reveals two ways to recover $x$:

1. sample $x_1 \sim \mathcal{N}(0, I)$ and follow the denoising dynamics;

2. apply the one-step map $x = x_1 - v^*(x_1, 1)$ directly.

3. combine 1 and 2: run the denoising dynamics until time $\tau$, then perform the final reconstruction in one step using $x = x_\tau - \tau\, v^*(x_\tau, \tau)$.

With a perfectly learned vector field, all choices recover the same $x$. In practice, however, the switching time $\tau$ becomes a meaningful control knob, leading to a distortion-perception trade-off.

On one hand, the perceptual realism typically improves monotonically as we switch later, since a later switch better leverages the pretrained model's generative ability. On the other hand, the distortion is typically non-monotonic w.r.t the switching time. Switching too late increases distortion due to error accumulation, while switching too early also lead to higher distortion as the sample is still dominated by noise and the learned vector field has larger error.

This non-monotonic distortion trend is characterized by the following proposition, which bounds the reconstruction error w.r.t the switching time $\tau$.

**Proposition B.1.** *Assume the learned vector field $v(\cdot, t)$ is $L$-Lipschitz for any $t$. Let $e_t$ denote the approximation error of the learned vector field at time $t$ along the clean path, and let $\delta_t$ be the state error at time $t$. Then the final reconstruction error $\delta$ satisfies*

$$\|\delta\| \le (\tau L + 1) \exp(L(1 - \tau)) \int_\tau^1 \|e_t\| \mathrm{d}t + \tau \|e_\tau\| \tag{12}$$

We derive this bound in Appendix D.2. In practice, we found $\|e_t\|$ increases as $t$ grows. From this bound, when $\tau \to 0$, the error is dominated by the integral over $e_t$, while when $\tau \to 1$, the error directly arises from the $e_1$.

We empirically demonstrate this trade-off on the 7 videos from UVG dataset, trained with $\lambda = 1.5e - 3$. We run inference with ODE discretized into 100 steps, early stop at every time step, and calculate PSNR, DISTS, LPIPS. In Figure 8, we visualize the metrics against the early stop time $\tau$. Note that the generation starts from $t = 1$ and ends at $t = 0$ when interpreting the plot. As shown in Figure 6, stopping early yields an essential gain of about 1.5 dB in PSNR. However, this comes at the expense of perceptual quality: perceptual metrics degrade noticeably, resulting in a clear distortion-perception trade-off.

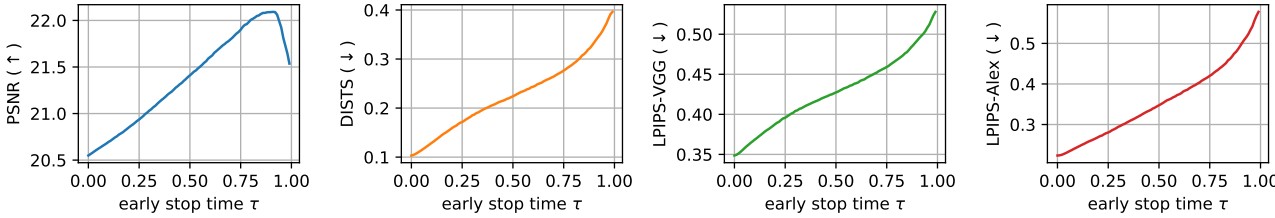

*Figure 8.* Distortion-Perception Trade-off via early stopping on UVG.

## B.2. Distortion-Perception Trade-off via Early Stopping *with Inference-time Scaling*

The above conclusion and theory are observed and derived for our method without inference-time scaling. Therefore, a natural question is whether the DP trade-off still holds when coupled with the inference-scaling approach we proposed. We investigate this on the same UVG videos as in the last section. We perform inference-time scaling with 1,000 steps, each with $2^{10}$ samples in Figure 9. As we can see, both distortion and perception metrics have the same trend. And early stop still provide a trade-off between distortion and perception when $t$ is smaller than a certain threshold. However, this threshold is much smaller than the case without inference-time scaling. This is because scaling corrects the error at the early stage of denoising ($t$ close to 1), leading to a smaller error accumulation.

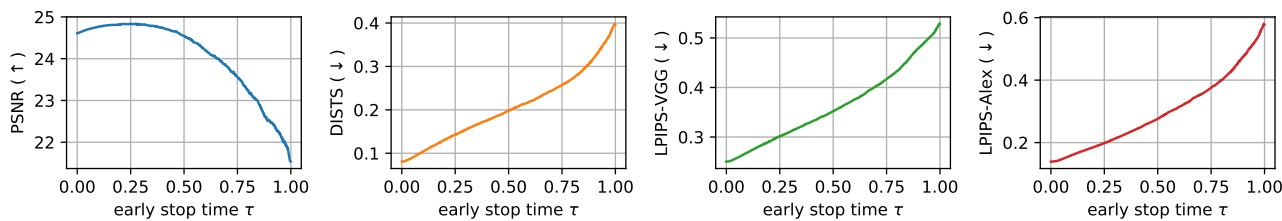

*Figure 9.* Distortion-Perception Trade-off via early stopping coupled with inference-time scaling on UVG.

## B.3. Comparison with INR Baseline

We compare VOV with a standard INR-based codec NVRC (Kwan et al., 2024). We use the code of NVRC available at `https://github.com/hmkx/NVRC`.

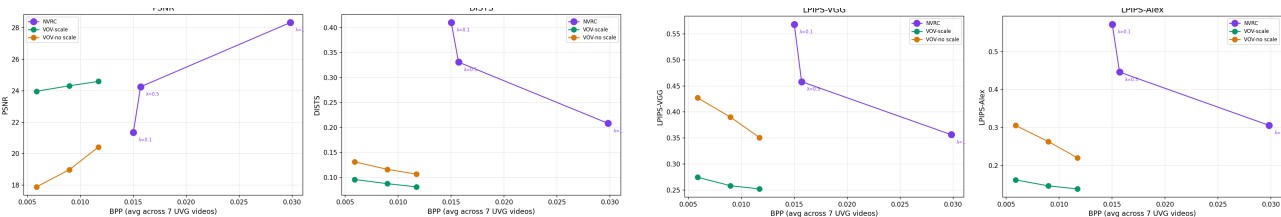

*Figure 10.* Rate-distortion comparisons with the previous INR-based video codec NVRC on (Kwan et al., 2024) on UVG.

## B.4. Pure Inference-time Scaling

We proposed a lightweight scaling procedure on top of the one-vector code to further improve compression performance. When allowing a more expensive scaling procedure, one can instead apply scaling directly on top of the original pretrained diffusion model, and use it to encode the target sample directly. In this section, we study the performance of this strategy.

In Figure 11, we report the compression performance of this direct scaling approach. Here, $\lambda = \infty$ corresponds to using no one-vector adaptation, while other values of $\lambda$ correspond to results with one-vector adaptation, trained with different RD trade-offs. As shown in the figure, this pure scaling strategy can, in fact, achieve more efficient compression performance. This is not surprising: direct scaling treats the pretrained model as a black-box generative prior, whereas our one-vector approach needs to capture correlations between adapted parameters in order to obtain a compact representation.

However, achieving strong performance with direct scaling requires a larger number of denoising steps (4000-10000), making both encoding and decoding impractically expensive. Moreover, because this scaling procedure only encodes the sample index, it does not provide a continuous representation of the sample. In contrast, the one-vector representation gives a flexible functional representation of the sample, which naturally supports editing, further scaling, and control. It also opens a promising future direction: learning an encoder and decoder for the representation, potentially enabling faster coding and more compact amortized compression.

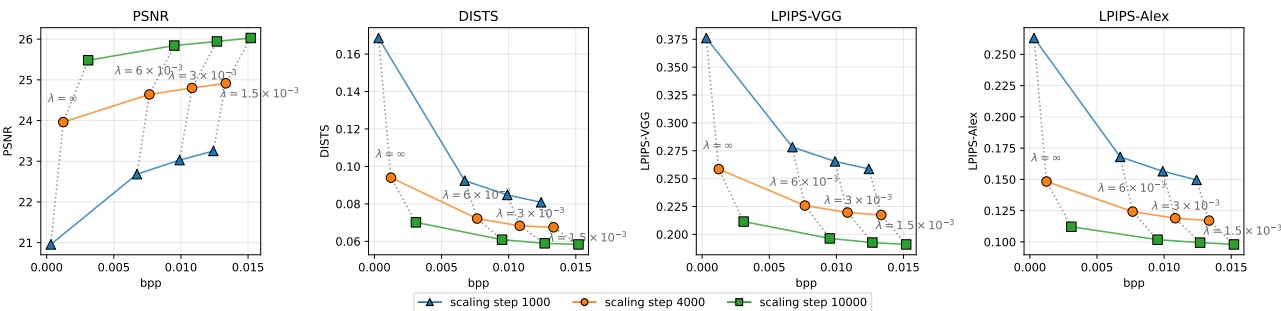

*Figure 11.* Compression performance of direct scaling on the pretrained diffusion model at different bitrates and scaling budgets. Here, $\lambda = \infty$ denotes the setting without one-vector adaptation.

## B.5. Additional Results on Editing

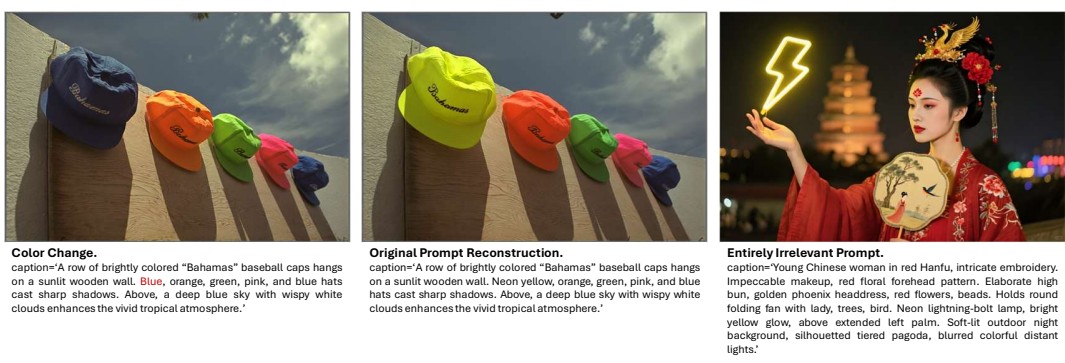

**Color Change.**
caption='A row of brightly colored "Bahamas" baseball caps hangs on a sunlit wooden wall. Blue, orange, green, pink, and blue hats cast sharp shadows. Above, a deep blue sky with wispy white clouds enhances the vivid tropical atmosphere.'

**Original Prompt Reconstruction.**
caption='A row of brightly colored "Bahamas" baseball caps hangs on a sunlit wooden wall. Neon yellow, orange, green, pink, and blue hats cast sharp shadows. Above, a deep blue sky with wispy white clouds enhances the vivid tropical atmosphere.'

**Entirely Irrelevant Prompt.**
caption='Young Chinese woman in red Hanfu, intricate embroidery. Impeccable makeup, red floral forehead pattern. Elaborate high bun, golden phoenix headdress, red flowers, beads. Holds round folding fan with lady, trees, bird. Neon lightning-bolt lamp, bright yellow glow, above extended left palm. Soft-lit outdoor night background, silhouetted tiered pagoda, blurred colorful distant lights.'

*Figure 12.* Image editing results with one-vector representation. In the last example, we also show that image generation ability is largely maintained.

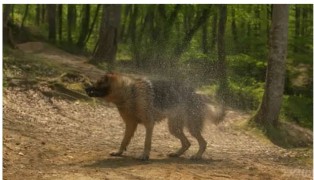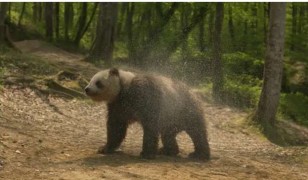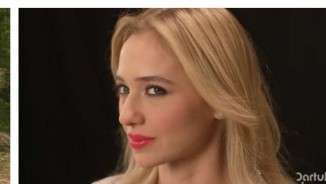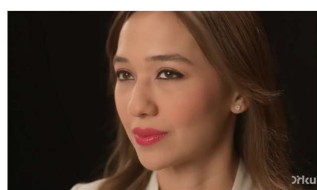

caption='In a sunlit forest clearing, a large German Shepherd stands on a dirt path, soaked and vigorously shaking off water. Droplets spray in a shimmering halo around its body, catching the warm light between the tall trees. The dog's fur ripples and its head whips from side to side as it braces its legs on the uneven ground, capturing a playful, energetic moment of relief after getting wet during an outdoor adventure.'

caption='In a sunlit forest clearing, a large panda stands on a dirt path, soaked and vigorously shaking off water. Droplets spray in a shimmering halo around its body, catching the warm light between the tall trees. The panda's fur ripples and its head whips from side to side as it braces its legs on the uneven ground, capturing a playful, energetic moment of relief after getting wet during an outdoor adventure.'

caption=In a dark studio setting, a woman with long, wavy blonde hair is filmed in tight close-up, wearing sparkling stud earrings. Her hair gradually lifts and whips around her head as if blown by a strong fan, creating strands that fly sideways and upward against the black background. The lighting remains soft and even throughout, emphasizing the movement of her hair while she otherwise stays relatively still, suggesting a stylized beauty or promotional shoot.'

caption='In a dark studio setting, a woman with long, wavy black hair is filmed in tight close-up, wearing sparkling stud earrings. Her hair gradually lifts and whips around her head as if blown by a strong fan, creating strands that fly sideways and upward against the black background. The lighting remains soft and even throughout, emphasizing the movement of her hair while she otherwise stays relatively still, suggesting a stylized beauty or promotional shoot.'

*Figure 13.* Video editing results on Wan-2.1-14B, with one vector of length 524288. In the first example, changing the dog to a panda results in a coherent edit. However, in the second example, due to biases in the base model, black hair is often associated with Asian individuals, which leads the model to modify the face accordingly.

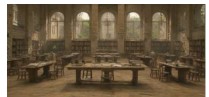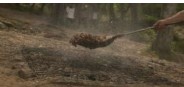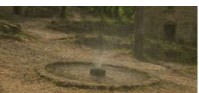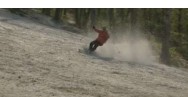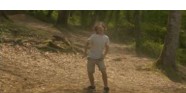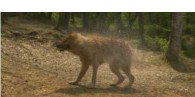

**Entirely Irrelevant.**
caption='Inside a quiet library reading room, long wooden tables are arranged in neat rows beneath tall arched windows. Soft daylight enters through the glass and spreads evenly across the polished surfaces of the desks. Shelves filled with books line the walls from floor to ceiling, their spines forming dense patterns of muted colors and worn textures. A few open books rest on the tables beside neatly stacked notebooks and reading lamps. The atmosphere is calm and orderly, with the stillness of the room emphasizing the quiet space for reading and study.'

**Almost Irrelevant.**
caption='In a crowded open-air market at dusk, a street vendor flips a large metal pan filled with roasted chestnuts over a charcoal brazier. As the pan jerks upward, the chestnuts tumble and roll against one another, sending tiny flakes of charred shell and sparks from the glowing coals drifting into the cool evening air. Thin ribbons of fragrant smoke rise from the brazier, twisting upward and catching the orange glow of hanging lanterns. The vendor's hands move quickly as the pan settles back onto the grill, stirring the chestnuts with a metal scoop that scrapes loudly against the pan. Small bursts of ember light flicker beneath the grate, illuminating the glossy shells as they shift and crackle. The moment captures the warmth and motion of street cooking, where smoke, sparks, and tumbling chestnuts create a lively scene in the fading evening light.'

**Only Similar in Vibe.**
caption='In a sunlit courtyard between old stone buildings, a narrow gravel path runs across the open space while warm afternoon light falls between the surrounding walls. Near the center of the courtyard, a wooden fountain slowly spills clear water into a shallow stone basin. Small streams of water arc gently outward and break into fine droplets that glint briefly in the sunlight before landing on the wet stone surface. Ivy climbs along the nearby walls, and patches of moss grow between the uneven stones around the basin. The scene captures a quiet outdoor moment where light, water, and the open space of the courtyard create a calm atmosphere during a warm afternoon.'

**Contain Similar Motion Pattern.**
caption='On a snowy mountain slope under a pale winter sun, a skier in a bright red jacket carves sharply across the powder. As the skis slice through the fresh snow, a wide plume of fine white powder erupts behind them, sparkling in the cold light. The skier leans deeply into the turn, knees bent and arms steady, sending an arc of snow crystals into the air that drift and swirl before settling back onto the slope. The crisp wind lifts loose flakes from the surface, and the skier's movement leaves a smooth curved trail etched into the untouched snow. In the background, tall pine trees stand dusted with frost, their branches glinting faintly as the airborne powder catches the sunlight. The moment captures the fluid motion and exhilaration of a fast downhill carve, with the snow cloud expanding and dissolving into the bright mountain air.'

**Contain Only Background.**
caption='In a sunlit forest clearing on a dirt path. Warm light filters between the tall trees. The moment captures a playful, energetic scene during an outdoor adventure.'

**Contain Key Object.**
caption='In a grassy backyard just after a summer rainstorm, a large dog charges across the wet lawn chasing a bright yellow tennis ball. As the dog skids to a sudden stop, its paws dig into the soft ground and send small clumps of damp grass and droplets of water flying into the air. The dog grabs the ball and immediately begins to shake its head excitedly, whipping its ears from side to side while beads of water spray outward in tiny sparkling arcs. Its thick fur ripples with the motion as the dog braces its legs against the slippery grass. Behind it, shallow puddles shimmer in the fading sunlight and blades of grass bend under the scattered droplets. The moment captures the energetic burst of movement and joy as the dog bounds across the yard, shaking off rainwater and scattering glistening droplets through the warm evening light.'

*Figure 14.* Overfitting in the representation can also influence the generation under irrelevant prompts. When the prompt is highly relevant to the scene or contains key objectives, the model's output is strongly influenced and may even directly reproduce the overfitted video. For prompts that are less relevant, the influence is weaker but still observable, affecting attributes such as color, motion, or background.

## B.6. Trade-off between Generation and Memorization

Beyond reconstruction, implicit visual representations can remember the sample and influence the generation behavior of the underlying diffusion foundation model. In practice, we observe that the number of fine-tuning iterations plays a critical role in balancing faithful memorization of a given visual signal and preservation of the model's general generative capability.

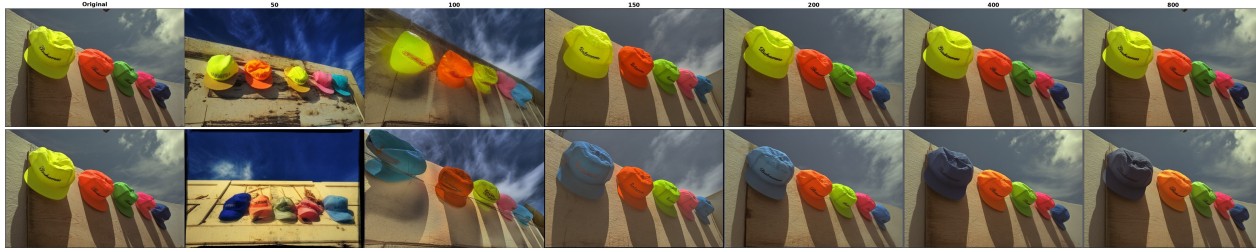

*Figure 15.* Above: Reconstruction with original image captions "A row of brightly colored 'Bahamas' baseball caps hangs on a sunlit wooden wall. Neon yellow, orange, green, pink, and blue hats cast sharp shadows. Above, a deep blue sky with wispy white clouds enhances the vivid tropical atmosphere.". Below: Editing with image captions "A row of brightly colored 'Bahamas' baseball caps hangs on a sunlit wooden wall. Blue, orange, green, pink, and blue hats cast sharp shadows. Above, a deep blue sky with wispy white clouds enhances the vivid tropical atmosphere."

*Figure 16.* Overfitting prompt: "A lone rockhopper penguin stands on rugged, dark gray coastal rocks, wings slightly outstretched. Its black-and-white body contrasts with bright yellow head crests and pink beak and feet, creating a crisp, windswept, subantarctic shoreline atmosphere." Generation prompt: "A lone penguin."

Specifically, as the model becomes increasingly overfitted to the target image, samples generated from partially related prompts become more similar to that overfitted image. We provide examples in Figure 15 and Figure 16, including the reconstructed images and the edited images in terms of different overfitting iterations. As we can see, more overfitting iterations generally improve the editing and reconstruction results. However, it will also influence the generation with different prompts (Figure 16).

We further find that this trade-off between fitting iterations and memory capability is strongly dependent on the capacity of the underlying foundation model. Larger and more expressive models (e.g., Qwen-Image-20B (Wu et al., 2025)) exhibit higher resilience to adaptation, requiring substantially more optimization steps (often exceeding 500 iterations) before noticeable degradation of generative ability occurs. In contrast, smaller or weaker models are significantly more fragile: even moderate levels of overfitting can quickly disrupt their original generation behaviour, and highly irrelevant prompts can also be influenced (Wan-2.1-14B in Figure 14.)

### B.7. Catastrophic Mixing from Visual Memory

As an extension to Figure 7, we also tried to learn a single group of LoRA weights to represent 10 images, where the content of these images are relevant in 5 pairs. As shown in Figure 17, there is a "Catastrophic Mixing" phenomenon observed: the reconstructed images bleed into each other, because the prompt of entities (e.g., the penguin) got mixed connections to visual content. This suggests that the similarity between prompts/objectives appears to play an important role in LoRA-based visual memories. In contrast, if the contents of 10 images are irrelevant, the reconstructed images are much better in reconstruction quality (shown in Figure 18).

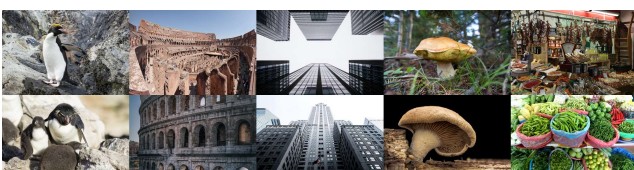 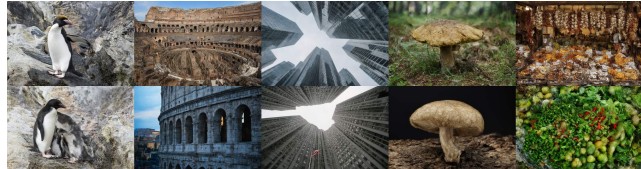

*Figure 17.* Left: Original images that are used for learning a single group of implicit visual representations. Right: Reconstructed images from the learned single group of implicit visual representations. The content in two rows of images are relevant.

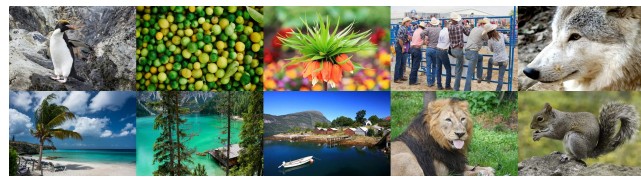 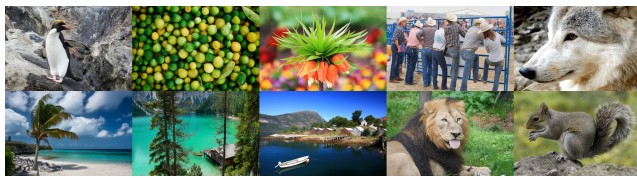

*Figure 18.* Left: Original images that are used for learning a single group of implicit visual representations. Right: Reconstructed images from the learned single group of implicit visual representations. The content in all ten images are irrelevant.

# C. Additional Backgrounds and Related Works

## C.1. Explicit + Implicit Representations for Visual Compression

Recently, several works have explored combining explicit and implicit representations for visual compression, aiming to balance entropy efficiency and expressive reconstruction. For images, COOL-CHIC (Ladune et al., 2023) proposes a hybrid codec that explicitly models entropy with an autoregressive prior while decoding pixels using an implicit coordinate-based neural representation. For videos, HNeRV (Chen et al., 2023) extends the NeRV framework by hierarchically encoding video content into explicit network parameters and latent codes, while relying on an implicit neural function to reconstruct frames from continuous coordinates. Later, NVRC (Kwan et al., 2024) introduces a neural video representation that combines explicitly quantized latent variables for entropy coding with an implicit neural decoder for frame synthesis. These works can be regarded as a combination of explicit compression and implicit compression, which are shown in Figure 1a and 1b.

## C.2. LoRA as Visual Memory

Personalization and controllable adaptation both full-parameter and parameter-efficient fine-tuning. DreamBooth (Ruiz et al., 2023) demonstrates that a diffusion model can memorize a subject or visual concept by overfitting on a small set of images, effectively treating model parameters as a form of visual memory. However, this requires updating a large portion of the model and incurs significant storage and deployment cost per concept. In the video domain, methods such as Customize-A-Video (Ren et al., 2024) and VideoMage (Huang et al., 2025) show that LoRA modules can encode motion or subject-specific priors from limited examples and be reused across novel scenes, highlighting their role as transferable motion or identity memories. LoRA-Edit (Gao et al., 2025a) demonstrates that LoRA can record localized, edit-specific behaviours that persist temporally while preserving the frozen backbone. Beyond single-adapter usage, contrastive test-time composition of multiple LoRAs reveals that independently trained adapters can be combined at inference to retrieve and blend multiple visual concepts, reinforcing a memory-bank interpretation (Meral et al., 2025). At the system level, V-LoRA (Mi et al., 2025) formalizes LoRA as an on-demand, deployable unit for large language models. Collectively, these works suggest that LoRA adapters function not merely as optimization tools, but as compact implicit visual memories that can be stored, composed, and injected into foundation models to control generation behavior.

## C.3. Diffusion and Flow Matching

In this section, we provide more background on diffusion models (Song et al., 2021) and flow matching (Lipman et al., 2023), which are equivalent in certain cases.

**Diffusion models.** Diffusion models define a forward noising SDE, corrupting data into noise:

$$\mathrm{d}x_t = \beta_t x_t \mathrm{d}t + \sigma_t \mathrm{d}\overrightarrow{w}_t, \quad x_0 \sim p_{\mathrm{data}} \tag{13}$$

$\beta_t$ and $\sigma_t$ hyperparameters, which are typically chosen so that the final state of this SDE follows Gaussian distribution. For simplicity, without loss of generality, we now assume $t \in [0, 1]$ and $\beta_t$ and $\sigma_t$ are chosen such that $x_1 \sim \mathcal{N}(0, 1)$.

Assume the marginal density of the SDE at time step $t$ is $p_t$, the score function is defined as $\nabla \log p_t$. By Nelson's relation (Nelson, 1967; Anderson, 1982), the time-reveral of the above SDE is given by

$$\mathrm{d}x_t = \left( \beta_t x_t - \sigma_t^2 \nabla \log p_t(x_t) \right) \mathrm{d}t + \sigma_t \mathrm{d}\overleftarrow{w}_t, \quad x_1 \sim \mathcal{N}(0, 1) \tag{14}$$

The arrow indicates the direction of the process. Therefore, once we define the forward SDE, we only need to find the score function to reverse it. In practice, we train a network $s_\theta(x_t, t)$ to approximate the score function by denoising score matching (DSM, Vincent, 2011):

$$\mathbb{E}_{x \sim p_{\mathrm{data}}, \epsilon \sim \mathcal{N}(0,1)} \left[ \lambda(t) \| s_\theta(x_t, t) - \nabla \log p(x_t | x_0 = x) \|^2 \right] \tag{15}$$

where $\lambda(t)$ is an optional weighting function, and $x_t | x_0 = x$ is defined by the forward SDE. In fact, the forward SDE can be written as a Gaussian kernel:

$$p(x_t | x_0 = x) = \mathcal{N}(x_t | m_t(x), v_t) \tag{16}$$

$$m_t(x) = \alpha_t x, \quad \alpha_t = \exp\left( \int_0^t \beta_s \mathrm{d}s \right), \quad v_t = \alpha_t^2 \int_0^t \frac{\sigma_s^2}{\alpha_s^2} \mathrm{d}s \tag{17}$$

Hence, when training with DSM, we can directly calculate $x_t = \alpha_t x + \sqrt{v_t}\epsilon$, and $\nabla \log p(x_t|x_0 = x)$ is also analytically defined as the Gaussian kernel.

**Flow matching.** Flow matching seeks to find a vector field $v_\theta(x_t, t)$ that transport between the data and the Gaussian noise[2]. The vector field defines an ODE:

$$\mathrm{d}x_t = v_\theta(x_t, t)\mathrm{d}t \tag{18}$$

Note that, unlike the SDE that needs different formulation for te forward and backward process, the ODE is already bidirectional. To train the vector field, the most commonly choice is the conditional flow matching objective (Lipman et al., 2023). Concretely, we first construct an interpolant between $x_0$ and $x_1$, denoted by $x_t$, and then mininze

$$\mathbb{E}_{x \sim p_{\mathrm{data}}, \epsilon \sim \mathcal{N}(0,1)}\left[\|v_\theta(x_t, t) - \partial_t x_t\|^2\right] \tag{19}$$

For the mostly used linear interpolant, we have

$$x_t = (1 - t)x_0 + t\epsilon, \quad \partial_t x_t = \epsilon - x_0 \tag{20}$$

**Conversion between ODE and SDE** Assuming accessing the score function, one can convert the ODE to the SDE. Assume the marginal density for the above ODE is $\log p_t$, then, the following two SDE has the same marginal density as the ODE:

$$\mathrm{d}x_t = \left(v_\theta(x_t, t) + \frac{1}{2}\sigma_t^2 \nabla \log p_t(x_t)\right)\mathrm{d}t + \sigma_t \mathrm{d}\overrightarrow{w}_t, \quad x_0 \sim p_0 \tag{21}$$

$$\mathrm{d}x_t = \left(v_\theta(x_t, t) - \frac{1}{2}\sigma_t^2 \nabla \log p_t(x_t)\right)\mathrm{d}t + \sigma_t \mathrm{d}\overleftarrow{w}_t, \quad x_1 \sim p_1 \tag{22}$$

This is valid for any $\sigma_t > 0$. There are models directly training the vector field and score separately, known as stochastic interpolant (Albergo et al., 2025).

**Equivalent between flow matching and diffusion model.** Let's recall the flow matching objective and the DSM objective, one can realize that the learned vector field and the score network are both a linear function of $x_t$, and $\mathbb{E}[x_0|x_t]$. Therefore, one can derive the equivalence between the vector field and the score function. Taking the linear interpolant $x_t = (1 - t)x_0 + t\epsilon$ as an example, when the vector field is trained to the optimal solution, we have

$$\nabla \log p_t(x_t) = -\frac{x_t + (1 - t)v_\theta(x_t, t)}{t} \tag{23}$$

Therefore, we can write Equations (21) and (22) into SDEs where the drifts are only given by the score function and $x_t$.

Further, recall that we can choose $\sigma_t$ in Equations (21) and (22) freely. Therefore, we can set it so that the drift of the forward SDE in Equation (21) does not contains the score function. Precisely,

$$\sigma_t = \sqrt{2t/(1 - t)} \tag{24}$$

Then, the forward SDE will become

$$\mathrm{d}x_t = -\frac{1}{1 - t}x_t \mathrm{d}t + \sqrt{\frac{2t}{1 - t}}\mathrm{d}\overrightarrow{w}_t \tag{25}$$

In other word, a flow matching trained with linear interpolant and independent coupling, is equivalent to a diffusion model defined via the forward SDE in Equation (13), where

$$\beta_t = -\frac{1}{1 - t}, \quad \sigma_t = \sqrt{\frac{2t}{1 - t}} \tag{26}$$

---

[2]Flow matching can be broadly defined between any two distributions, with arbitrary couplings. However, for simplicity, this section discuss the flow matching between Gaussian noise and data distribution, and assume the coupling between them is dependent.

### C.4. Inference-time Scaling of Diffusion Models

Inference-time scaling aims to spend more compute in inference time to improve or steer the generation process of the diffusion models. A common framework is, given a pretrained diffusion model and a verifier, one runs several trajectories, or branch the trajectory with multiple particles, and then select the best particle according to the verifier. The framework that is the most relevant to our approach is those based on Sequential Monte Carlo (Wu et al., 2023), also known as Feynman-Kac method (Singhal et al., 2025) in a broader sense.

These approaches aim to to tilt the diffusion process towards a reward function. Specifically, for a pretrained diffusion model with denoising kernel $p(x_{t_{n-1}}|x_{t_n})$, assuming its final sample follow a distribution with density $p_0$, these approaches aim to draw sample from $p_0(x) \exp(r(x))$ where $r$ is a reward function, or a likelihood in a inverse problem.

First, these approaches define a proposal process, where the transition kernel is given by $q(x_{t_{n-1}}|x_{t_n})$. This process can be chosen to be the same as the pretrained diffusion, or can be guided by some guidance, similar to DPS (Chung et al., 2023). Then, they define a sequence of intermediate reward functions $r_{t_n}$. This can be selected to be the final reward $r$ evaluated on $\mathbb{E}[x_0|x_{t_n}]$, obtained following the Tweedie's formula (Efron, 2011), i.e.,

$$r_{t_n}(x_t) = \gamma_{t_n} r(\mathbb{E}[x_0|x_{t_n}]) \tag{27}$$

where $\gamma_{t_n}$ is a scaling factor and typically $\gamma_1 = 0$, and $\gamma_0 = 1$. Then, the SMC weight for each step is defined as

$$w \propto \frac{p(x_{t_{n-1}}|x_{t_n}) \exp(r_{t_{n-1}}(x_{t_{n-1}}))}{q(x_{t_{n-1}}|x_{t_n}) \exp(r_{t_n}(x_{t_n}))} \tag{28}$$

More broadly, similar SMC algorithm can be used for inference-time annealing, product, or CFG debiasing, for both Gaussian diffusion models and discrete diffusions (Skreta et al., 2025; He et al., 2026; Lee et al., 2025; Ren et al., 2026).

From this perspective, our proposed scaling algorithm takes a trivial reward function $r = 0$, use the adapted diffusion model as the prior, and take the analytical kernel in Equation (9) as the target. In principle, we could further guide the generation by introducing a reward and performing reward-guided selection. However, doing so would substantially increase the number of function evaluations (NFE). In contrast, our current algorithm introduces no additional functional calls: it only adds Gaussian-kernel sampling and evaluation, which is lightweight relatively.

### C.5. Reverse Channel Coding and Relative Entropy Coding

Reverse Channel Coding, also known as Relative Entropy Coding, aims to encode a random sample from a target distribution $q$, using a shared prior distribution $p$ and the shared PRNG. For a more comprehensive introduction and explanation, we refer the reader to (Theis & Ahmed, 2022).

The most intuitive approach for performing relative entropy coding is minimal random coding (MRC, Havasi et al., 2019), where the encoder draws $M$ samples from $p$ using the shared PRNG, and transmit the index of the selected sample to the decoder. Here, to ensure the sample has small bias, $M$ is typically set to $\exp(D_{\mathrm{KL}}[q||p])$, and encoding the index costs roughly $D_{\mathrm{KL}}[q||p]$ nats. Several following approaches improve the coding efficiency of MRC. Particular, ordered random coding (ORC, Theis & Ahmed, 2022) added a permuted Gumbel noise to ensure that we not only choose the best sample, but also choose the one with the smallest index. A* Coding (Flamich et al., 2022) and Greedy Poisson rejection sampling (GPRS, Flamich, 2023) utilize Poisson process to control the bitrate in a similar way as ORC. Due to the Poisson process, ORC, A* Coding and GPRS could achieve a finite codelength even when for infinite number of candidates drawn from $p$. One can show that the theoretical codelength of these approaches is bounded by

$$D_{\mathrm{KL}}[q||p] + \log_2(D_{\mathrm{KL}}[q||p] + 1) + \mathrm{const.} \tag{29}$$

where the KL is in bits.

However, Relative Entropy Coding is challenging. To ensure a small bias, these approaches all requires $\mathcal{O}(\exp(D_{\mathrm{KL}}[q||p]))$ or even $\mathcal{O}(\exp(D_\infty[q||p]))$ number of candidate samples. While some approaches aim to accelerate the algorithm (Flamich et al., 2022; Flamich, 2023; He et al., 2024b), they are still limited by strong assumptions on the distribution. Therefore, these algorithms are still constraint in practice.

Fortunately, in our proposed inference-time scaling algorithm, the Relative Entropy Coding component is only used to refine the performance of the basic compression via adaption. Therefore, even when the KL divergence between the target and

proposal is large, we do not need an exponentially large number of particles. In other word, we use the Relative Entropy Coding algorithm only as a lightweight refinement component, significantly reducing its complexity.

### C.6. Lossy Compression with Diffusion via Relative Entropy Coding or Fixed Codebooks

As discussed in the main text, our scaling algorithm can also be interpreted through the lens of Diff-C (Theis et al., 2022), a lossy compression framework based on diffusion models and relative entropy coding. A key difference lies in computation required for Diff-C and our scaling algorithm. Diff-C uses an unadapted, generic diffusion model to define the proposal denoising kernel $p(x_{t_{n-1}}|x_{t_n})$. As a result, the number of candidates required for effective selection can become prohibitive—often scaling exponentially with the KL divergence between the target and the proposal.

Additionally, Diff-C encodes only the first $N' < N$ diffusion steps. The remaining steps are directly generated by the pretrained diffusion. To control the bitrate, Diff-C adjust $N'$. In contrast, our approach controls the bitrate primarily through the entropy model applied to the adaptation parameters. The inference-time scaling component contributes only a negligible overhead—approximately 0.00005–0.0005 bpp for our 480p video clips, and therefore we do not truncate the process as in Diff-C; instead, we apply it over the entire generation trajectory.

Recently, Vonderfecht & Liu (2025) introduced a CUDA-accelerated implementation of Diff-C, making Diff-C practical for the first time. Although our scaling algorithm is not as computationally demanding as Diff-C, it can also benefit from their implementation for further acceleration.

DDCM (Ohayon et al., 2025; Vaisman et al., 2026) proposed an alternative approach to leverage a pretrained diffusion model for compression using a fixed random codebook. This can be seen as a variation of Diff-C, where instead of using a randomly sampled index from the codebook, one takes the best candidate yielding the best inner product with the target. They also proposed an iterative approach to encode more than one candidate for larger bitrates.

While both DiffC and DDCM yield efficient compression by directly leveraging the diffusion model knowledge. Our approach is still substantially different in how the knowledge is used. DiffC and DDCM encode the noise candidate index or the codebook index at each step, whereas our approach aims to first learn a *continuous functional representation* by compressing the target into a parameter-space adaptation attached to the model. This can lead to important downstream differences. Our encoded representation, together with the base model, is a new diffusion model, which enables more flexible denoising procedures, including inference-time scaling or downstream editing. In contrast, Diff-C is closer to a modification of the inference algorithm rather than a parameterized adaptation of the model itself. The function that Diff-C or DDCM encodes is fixed and has less flexibility. Additionally, the index encoded in DiffC or DDCM is highly discontinuous and random, making it difficult to amortize. In contrast, our approach is compatible with a promising future direction: training an encoder to output the adaptation directly, potentially enabling a more compact representation and faster coding time.

## D. Proposition and Proofs

### D.1. Justification of the Overfitting Objective

In this section, we motivate our training objective in Equation (1) from the perspective of MDL, assuming the diffusion process. For easier reference, we will repeat key equations below.

The generation process of a pretrained diffusion model is written as

$$\mathrm{d}x_t = \left( \beta_t x_t - \sigma_t^2 \nabla \log p_t^{\mathrm{data}}(x_t) \right) \mathrm{d}t + \sigma_t \, \mathrm{d}\overleftarrow{w}_t, \qquad x_1 \sim \mathcal{N}(0, I), \tag{30}$$

where $p_t^{\mathrm{data}}$ denotes the density of the data distribution convolved with the Gaussian noise at time $t$ (i.e., the marginal distribution of $x_t$ under the forward noising process). Under the flow-matching parameterization, one has $\beta_t = -1/(1-t)$ and $\sigma_t = \sqrt{2t/(1-t)}$, and the score function is directly related to the learned vector field.

This SDE induces a path space together with a probability measure, which we denote by $\mathbb{P}$. Our goal is to construct a new measure $\mathbb{P}'$ that reconstruct $x$ at $t = 0$ while introducing as little additional information as possible relative to $\mathbb{P}$. The information-minimizing solution is the posterior of $\mathbb{P}$ given the terminal event $x_0 = x$, which can be characterized via Doob's $h$-transform.

Concretely, conditioning the base dynamics on the event $x_0 \in \mathcal{A}$ yields the controlled reverse-time SDE

$$\mathrm{d}x_t = \left(\beta_t x_t - \sigma_t^2 \nabla \log p_t^{\text{data}}(x_t) - \sigma_t^2 \nabla \log h_t(x_t)\right) \mathrm{d}t + \sigma_t \, \mathrm{d}\overleftarrow{w}_t, \qquad x_1 \sim \mathcal{N}(0, I), \tag{31}$$

where

$$h_t(x_t) = \mathbb{P}\{x_0 \in \mathcal{A} \mid x_t\} \tag{32}$$

is the probability of hitting $\mathcal{A}$ at $t = 0$ when starting from $x_t$ and following to the base dynamics. In the special case $\mathcal{A} = \{x\}$, $h_t(x_t)$ reduces to the conditional terminal density $p(x_0 = x | x_t)$.

Let's look at the term $\sigma_t^2 \nabla \log p_t^{\text{data}}(x_t) + \sigma_t^2 \nabla \log h_t(x_t)$, this can be simplified as

$$\sigma_t^2 \nabla \log p_t^{\text{data}}(x_t) + \sigma_t^2 \nabla \log h_t(x_t) \tag{33}$$

$$= \sigma_t^2 \nabla (\log p(x_0 = x | x_t) + \log p_t^{\text{data}}(x_t)) \tag{34}$$

$$= \sigma_t^2 \nabla (\log p(x_t | x_0 = x) + p_0^{\text{data}}(x_0 = x)) \tag{35}$$

$$= \sigma_t^2 \nabla \log p(x_t | x_0 = x) \tag{36}$$

The following equality holds as $\nabla$ is taken only over $x_t$, and the last line holds when the pretrained diffusion perfectly reverses the noising forward process, which is a common assumption we can take.

Finally, notice that $p(x_t | x_0 = x)$ is a simple Gaussian kernel, defined in Equation (16). Under the flow matching setup, this kernel is given by

$$p(x_t | x_0 = x) = \mathcal{N}(x_t | (1 - t)x_0, t^2) \tag{37}$$

Hence, the optimal vector field satisfies

$$-\frac{(x_t - (1 - t)x)}{t^2} = -\frac{x_t + (1 - t)v^*}{t} \tag{38}$$

Considering $x_t = (1 - t)x + t\epsilon$, we have

$$v^* = \epsilon - x \tag{39}$$

This also coincides with the optimal vector field between a delta distribution at $x$ and Gaussian.

### D.2. Bounds on the Distortion of Early Stopping

*Proof.* The proof has three parts. We first analyze the error introduced by the one-step map, and then we will analyze the error introduced by the integrating the vector field with error. Finally, we will put the error together, obtaining an upper bound on the overall error.

**(1) Error introduced by one-step map.** We assume $x_1 \sim \mathcal{N}(0, 1)$ and $x_0 = x$ is the data to be compressed. We train a vector field, and the optimal solution is given by Equation (8):

$$v^*(x_t, t) = (x_t - x)/t \tag{40}$$

Therefore, the one-step prediction of $x$ (which we denoted as $\tilde{x}$) from $\tau$ is given by

$$\tilde{x} = \tilde{x}_\tau - \tau v_\theta(\tilde{x}_\tau, \tau) \tag{41}$$

Note that we also take the error in $\tilde{x}_\tau$ into account. We can hence calculate the reconstruction error as

$$\delta = \tilde{x} - x \tag{42}$$

$$= \tilde{x}_\tau - \tau v_\theta(\tilde{x}_\tau, \tau) - x_\tau + \tau v^*(x_\tau, \tau) \tag{43}$$

$$= \delta_\tau - \tau(v_\theta(\tilde{x}_\tau, \tau) - v^*(x_\tau, \tau)) \tag{44}$$

$$= \delta_\tau - \tau(v_\theta(\tilde{x}_\tau, \tau) - (x_1 - x)) \tag{45}$$

where $\delta_\tau$ denotes the error at step $\tau$.

**(2) Error introduced by error in the vector field.** Now, let's look at how $\tilde{x}_\tau$ is generated. We start from $x_1$ and follow the ODE backwards in time:

$$\frac{\mathrm{d}\tilde{x}_t}{\mathrm{d}t} = v_\theta(\tilde{x}_t, t) \tag{46}$$

and recall the ground truth path is also known:

$$\frac{\mathrm{d}x_t}{\mathrm{d}t} = v^*(x_t, t) = x_1 - x \tag{47}$$

We can hence calculate the error

$$\frac{\mathrm{d}\delta_t}{\mathrm{d}t} = v_\theta(\tilde{x}_t, t) - (x_1 - x), \quad \delta_1 = 0 \tag{48}$$

and therefore,

$$\delta_t = \int_1^\tau (v_\theta(\tilde{x}_t, t) - (x_1 - x))\mathrm{d}t \tag{49}$$

$$= -\int_\tau^1 (v_\theta(\tilde{x}_t, t) - (x_1 - x))\mathrm{d}t \tag{50}$$

**(3) Putting them together.** Putting the errors together, we obtain

$$\|\delta\| = \|\int_\tau^1 (v_\theta(\tilde{x}_t, t) - (x_1 - x))\mathrm{d}t + \tau(v_\theta(\tilde{x}_\tau, \tau) - (x_1 - x))\| \tag{51}$$

Now, introduce the error of the learned vector field:

$$e_t = v_\theta(x_t, t) - (x_1 - x) \tag{52}$$

Note that the error is introduced on the clean path $x_t$ instead of $\tilde{x}_t$. Therefore, we have

$$\|v_\theta(\tilde{x}_t, t) - (x_1 - x)\| = \|v_\theta(x_t, t) - (x_1 - x) + v_\theta(\tilde{x}_t, t) - v_\theta(x_t, t)\| \tag{53}$$

$$\leq \|v_\theta(x_t, t) - (x_1 - x)\| + \|v_\theta(\tilde{x}_t, t) - v_\theta(x_t, t)\| \tag{54}$$

As we further assume the learned network is $L$-Lipschitz continuous, hence

$$\|v_\theta(\tilde{x}_t, t) - (x_1 - x)\| \leq \|v_\theta(x_t, t) - (x_1 - x)\| + L\|\tilde{x}_t - x_t\| \tag{55}$$

$$= \|e_t\| + L\|\delta_t\| \tag{56}$$

Therefore, we have

$$\|\delta_\tau\| \leq \int_\tau^1 \|v_\theta(\tilde{x}_t, t) - (x_1 - x)\|\mathrm{d}t \leq \int_\tau^1 (\|e_t\| + L\|\delta_t\|)\mathrm{d}t \tag{57}$$

With Grönwall's inequality (Gronwall, 1919), we have

$$\|\delta_\tau\| \leq \exp(L(1 - \tau)) \int_\tau^1 \|e_t\|\mathrm{d}t \tag{58}$$

Therefore, we have

$$\|\delta\| \leq (\tau L + 1)\|\delta_\tau\| + \tau\|e_\tau\| \leq (\tau L + 1)\exp(L(1 - \tau)) \int_\tau^1 \|e_t\|\mathrm{d}t + \tau\|e_\tau\| \tag{59}$$

$\square$

# E. Experimental Details

For image experiments, we use `Qwen-Image` (Wu et al., 2025) as our base model; and for video experiments, we use `Wan2.1-T2V-1.3B-Diffusers` (Wan et al., 2025) as our base model. We now describe additional experimental settings below.

## E.1. Representation Experiments

The detailed settings for experiments of image and video representations are described in Table 1 and 2.

| Training | |
| --- | --- |
| Image | Kodim03 from (Kodak, 1993) |
| Resolution | 768x512 |
| Generative model | Qwen-Image-20B (Wu et al., 2025) |
| Transformer number | 60 |
| LoRA rank $r$ | 1/2/4 |
| LoRA position | QKVO |
| One-vector size $k$ | $2^{15}$ / $2^{16}$ / $2^{17}$ / $2^{18}$ |
| Training iterations | 1000 |
| Optimizer | AdamW, weight decay 1e-4 |
| Learning rate | 0.0015 |
| Batch size | 64 |
| **Inference** | |
| Denoising steps | 50 |
| Classifier-free guidance | Disabled |

Table 1. Settings for LoRA- or one-vector image representations.

| Training | |
| --- | --- |
| Video | Beauty from (Mercat et al., 2020) |
| Resolution | Cropped and resized to 832x480x81 |
| Generative model | Wan-2.1-1.3B (Wan et al., 2025) |
| Transformer number | 30 |
| LoRA rank $r$ | 1/2/4/8 |
| LoRA position | QKVO |
| One-vector size $k$ | $2^{15}$ / $2^{16}$ / $2^{17}$ / $2^{18}$ |
| Training iterations | 700 |
| Optimizer | AdamW, weight decay 1e-4 |
| Learning rate | 0.002 |
| Batch size | 32 |
| **Inference** | |
| Denoising steps | 50 |
| Classifier-free guidance | Disabled |

Table 2. Settings for one-vector video representations.

## E.2. Compression Experiments

To compress the one-vector representation into bitstream, we adopt a two-stage training strategy. At the first stage, we train the one-vector representations without entropy constraint. Note that hashing mapping is normalized similar to (Li et al., 2025). Then, we load the pretrained vector and initialize the parameters entropy model to optimize them together. During that time, the training objective is aggregation of the diffusion loss and the rate loss, where a Lagrange multiplier $\lambda$ controls the trade-off. We describe other hyperparameters for video compression task in Table 3.

| Training | |
| --- | --- |
| Resolution | Cropped and resized to 832x480x81 |
| Generative model | Wan-2.1-1.3B (Wan et al., 2025) |
| Transformer number | 30 |
| LoRA rank $r$ | 1 |
| LoRA position | QKVO |
| One-vector size $k$ | $2^{17} = 131072$ |
| Rate loss weight $\lambda$ | [1.5e-3, 3e-3, 6e-3] |
| Training iterations | 1000 for stage-1, 1000 for stage-2 |
| Optimizer | AdamW, weight decay 1e-4 |
| Learning rate | 0.0015 |
| Batch size | 64 |
| **Inference** | |
| Denoising steps | 100 |
| **Inference-time Scaling** | |
| Denoising steps | 100/1000 |
| Sample per step | $2^{18}/2^{10}$ |

Table 3. Hyperparameters for compression and scaling experiments.

### E.3. Editing Experiments

Figure 7 in the main paper presents a collection of image editing experiments enabled by the proposed implicit visual representations. In the second row of the figure, the left-most image is generated by overfitting a LoRA-based representation to a single image (Kodim23 from the Kodak dataset (Kodak, 1993)), followed by sampling with a modified textual prompt. The second image is produced using the same procedure but with a different source image (Kodim07). The third image demonstrates *visual merging*: a single set of LoRA-based representations is jointly optimized over two source images (Kodim07 and Kodim01). During training, the source image within each mini-batch is randomly sampled from the image set, encouraging the learned representation to encode shared visual characteristics. The fourth image is obtained in the same manner but extends the collective representation to three source images. Finally, the right-most image illustrates resolution editing. After learning an implicit visual representation at a resolution of $768 \times 512$, the generation resolution is changed to $512 \times 768$ at inference time. Since the underlying diffusion foundation model supports variable-resolution generation, this experiment shows that the proposed implicit visual representations naturally inherit this capability, enabling flexible resolution adaptation without retraining. We emphasize that the editing experiment here is zero-shot, and the used base model originally does not accept native visual input.

### E.4. Configurations of Traditional Codecs

We used the reference configuration files recommended by standardization organizations. Specifically, for VTM, the cfg file is from `https://vcgit.hhi.fraunhofer.de/jvet/VVCSoftware_VTM/-/blob/master/cfg/encoder_randomaccess_vtm.cfg?ref_type=heads`; and for HM, we use the cfg file from `https://vcgit.hhi.fraunhofer.de/jvet/HM/-/blob/master/cfg/encoder_randomaccess_main10.cfg?ref_type=heads`. The used *Random Access* mode helps to produce the strongest performance from H.266/VTM or H.265/HM.

# F. Additional Results

## F.1. Encoding/Decoding Time

We report the encoding and decoding time of our method VOV (with & without scaling) in Table 4. We also compare with both traditional video codec and real-time neural video codec including DCVC-RT. The overfitting phase of VOV is measured on 8 NVIDIA A100 GPU in parallel, while others are measured on one NVIDIA A100 GPU. In our work, we do not explicitly optimize runtime. In addition, functional representations based on per-instance overfitting are inherently more time-consuming than conventional learned codecs. The runtime also scales with spatial resolution and video length, since the Diffusion Transformer requires more denoising steps as the number of visual tokens increases. As a promising future direction, we could develop an amortized encoder that directly predicts the LoRA adaptation, which could substantially reduce the encoding cost.

*Table 4.* Encoding and Decoding time.

| Method | Encoding per frame | Decoding per frame |
|---|---|---|
| VTM/H.266 RA mode | 30.51 s | 19.0 ms |
| DCVC-RT | 5.84 ms | 5.90 ms |
| Our VOV | ∼18 min (overfitting) | 2.48 s |
| + 1000-step scaling | +27 s (importance sampling) | 25 s |

## F.2. Full RD Curve

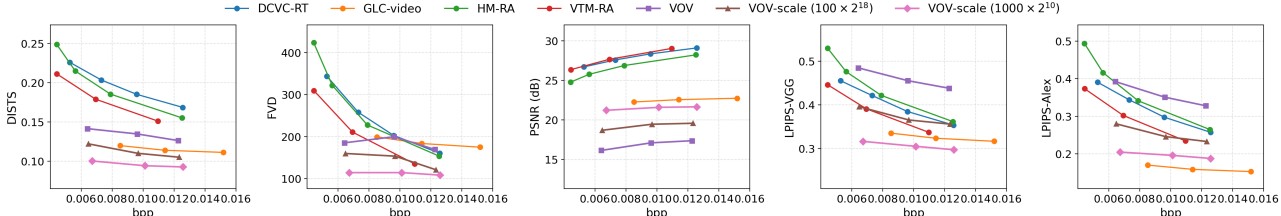

*Figure 19.* Comparisons with existing video codecs on HEVC-B. For DISTS, FVD and LPIPS, lower is better. For PSNR, higher is better.

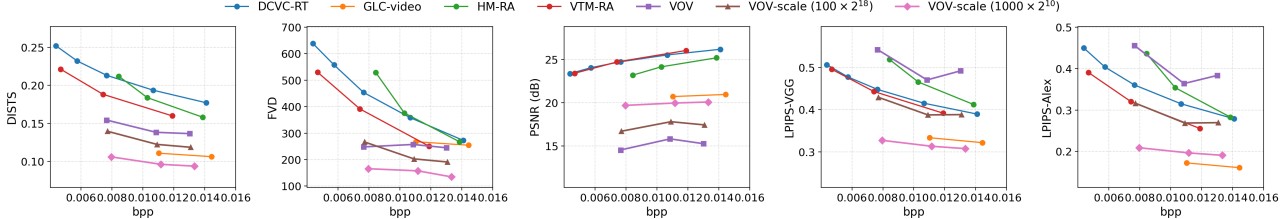

*Figure 20.* Comparisons with existing video codecs on HEVC-C. For DISTS, FVD and LPIPS, lower is better. For PSNR, higher is better.

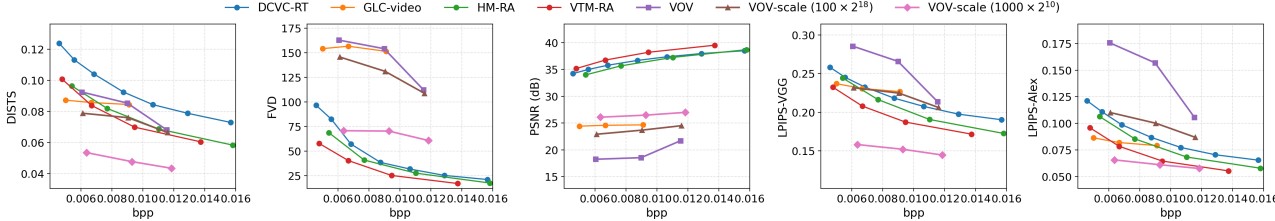

*Figure 21.* Comparisons with existing video codecs on HEVC-E. For DISTS, FVD and LPIPS, lower is better. For PSNR, higher is better.

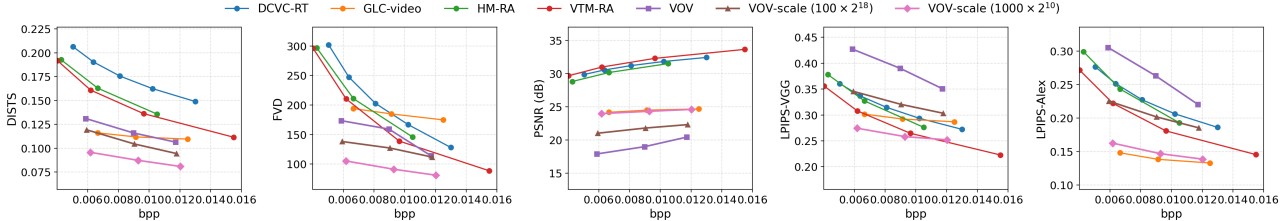

*Figure 22.* Comparisons with existing video codecs on UVG. For DISTS, FVD and LPIPS, lower is better. For PSNR, higher is better.

### F.3. Visualization of Compressed Videos

As shown in Figure 25, we provide more visualizations of decoded videos for qualitative comparison. The scaling results is with 1000 denoising steps, and $2^{10}$ samples per step. In addition, we save the decoded videos into mp4 files to supplementary material for better demonstration of the advantage of our method. It is shown that our method achieves much better perceptual quality, such as smooth temporal coherence, which cannot be reflected by rate-distortion curves.

Ground Truth

VTM
0.0211 bpp

DCVC-RT
0.0149 bpp

GLC-Video
0.0152 bpp

VOV
0.0126 bpp

VOV + Scaling
0.0129 bpp

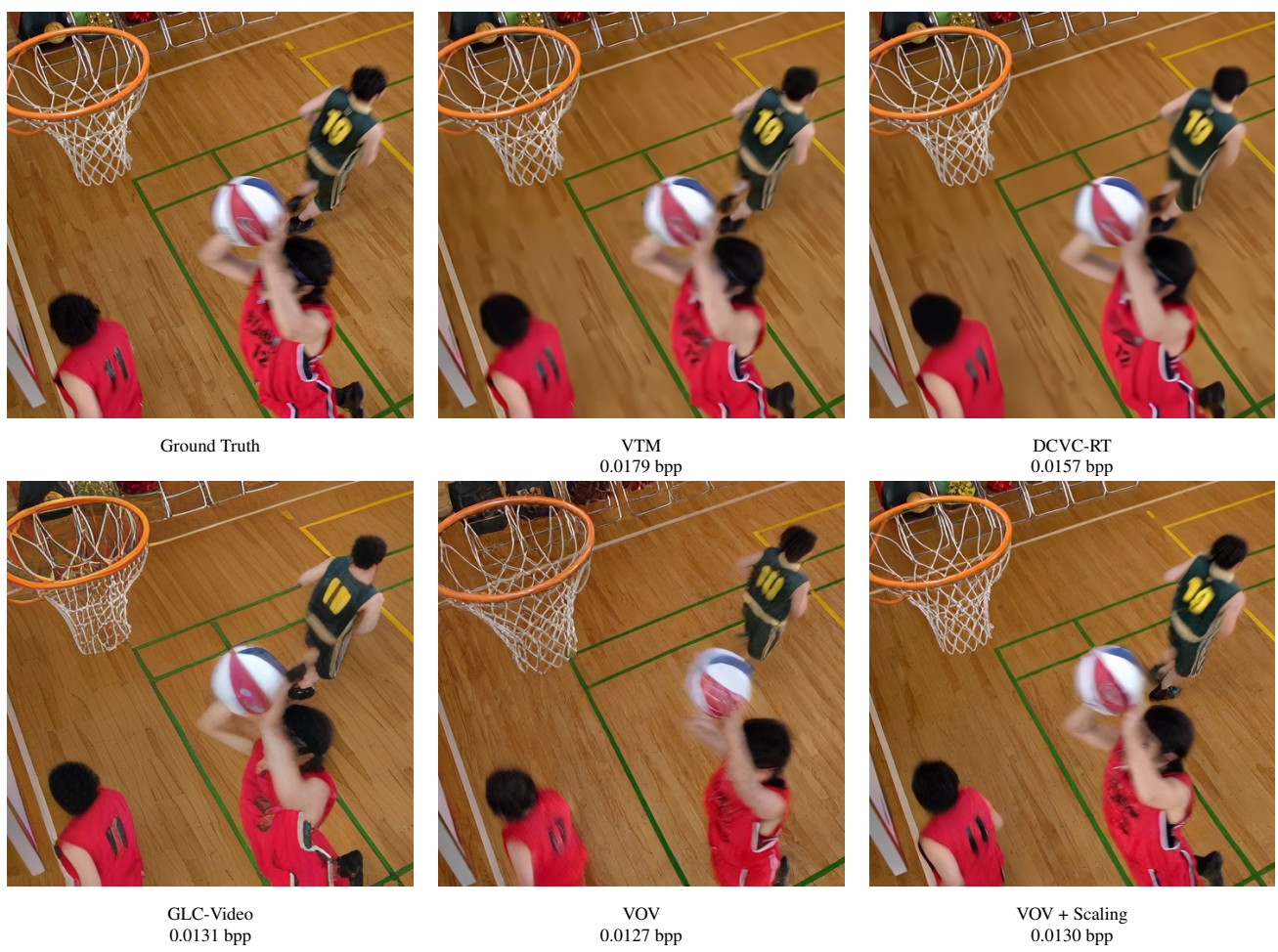

Ground Truth

VTM
0.0179 bpp

DCVC-RT
0.0157 bpp

GLC-Video
0.0131 bpp

VOV
0.0127 bpp

VOV + Scaling
0.0130 bpp

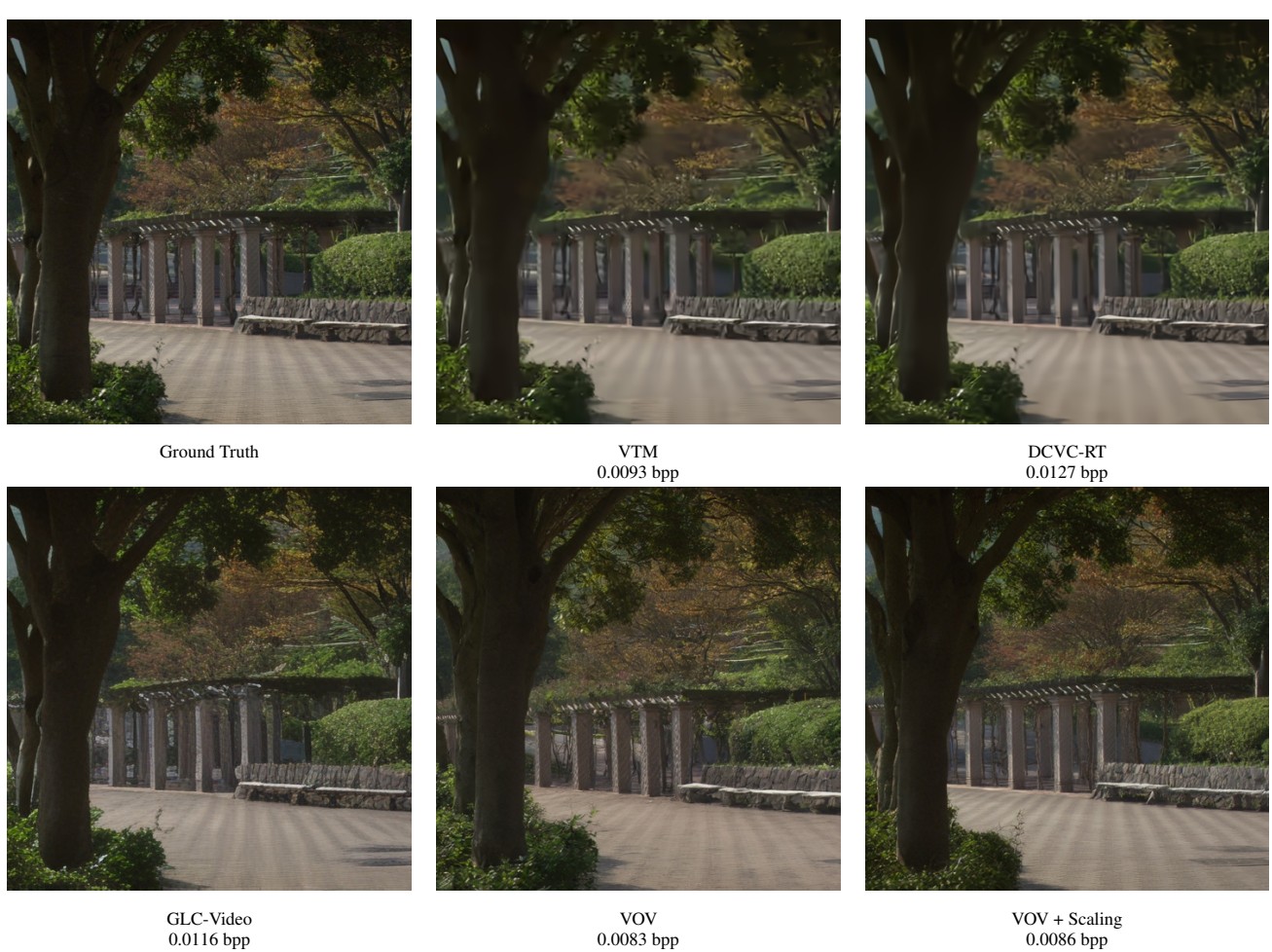

*Figure 25.* Additional visual comparisons with different video codecs.

