# OpenReview forum: "Compression as Adaptation: Implicit Visual Representation with Diffusion Foundation Models"
_ICML.cc/2026/Conference — ICML 2026 regular_

### Official Review · Reviewer_uXew · 2026-03-11

**Soundness:** 3
**Presentation:** 3
**Significance:** 3
**Originality:** 3
**Overall Recommendation:** 4
**Confidence:** 4

**Summary:**

This paper encodes an image/video as a LoRA adaptation attached to a frozen diffusion foundation model, then compresses the adaptation into a single hashed vector with quantization and entropy coding (VOV). It further uses SMC-style inference-time scaling/control to improve reconstruction with extra compute and small side information. Experiments target extremely low-bitrate video compression on UVG/HEVC at 480p, 81-frame clips, and also show prompt-based editing/composition for images.

**Compliance With Llm Reviewing Policy:**

Affirmed.

**Final Justification:**

The authors addressed my concerns during the rebuttal, I maintain my original positive score.

**Key Questions For Authors:**

1. What is the wall-clock encode time per 81-frame 480p clip (and decode time) for the main settings? How does this scale with resolution and clip length?

2. I find the Fig. 7 editing/composition results really interesting. In your experience, is this editability tied to the compression strength? For example, under stronger compression (lower bpp / smaller \(k\) / more aggressive quantization), does editing quality drop or become more stochastic/random? Also, Fig. 7 shows merging 2–3 images—what happens when you scale this to 5 or 10 images? Do you see “catastrophic mixing” where content starts to bleed into each other?

**Limitations:**

Yes

**Strengths And Weaknesses:**

## Strengths

1. Interesting problem: this work target at reusing a pretrained foundation model’s generative prior with minimal additional information to achieve strong compression. This problem is compelling and worth exploring.
2. Reasonable method design: The proposed VOV pipeline  is reasonable and coherent.
3. Nice extra capability: Using the learned LoRA as instance memory for personalized generation  is interesting beyond pure reconstruction.

## Weaknesses

1. Missing comparisons to INR/implicit-representation codecs: Given the “implicit/functional representation” framing, the experimental section should compare against representative INR-based compression methods.
2. No encode/decode runtime comparison reported: Please report wall-clock encoding/decoding time, ideally including a direct comparison to learned video codecs such as DCVC-RT / GLC-Video.

---

> ### Author Rebuttal · Authors · 2026-03-30
>
> Thank you for the constructive review. We appreciate that you found the problem compelling, the overall VOV pipeline coherent, and the learned LoRA representation interesting as an instance memory beyond pure reconstruction. We address your questions below.
>
> **1. Missing comparisons to INR / implicit-representation codecs.**  Following your suggestion, we have added an INR-based baseline (NVRC) on UVG dataset in https://anonymous.4open.science/r/rebuttal-E61D/nvrc_vov_metrics_vs_bpp.png.
>
> As shown, VOV achieves a stronger overall performance than NVRC. More importantly, while NVRC can obtain a relatively strong PSNR, its visual realism is substantially weak. We believe this reflects a key strength of our framework. Our functional representation does not operate in isolation; instead, it is coupled with a diffusion foundation model. This allows the method to maintain strong reconstruction fidelity while also producing outputs with significantly better realism.
>
>
> **2. Encode/decode runtime comparison.**
> Thank you for this excellent suggestion. We report the runtime comparison below. As also discussed in the limitations, our method currently incurs longer encoding time than existing baselines. We will make this point more explicit and include the table in the camera-ready version.
>
> | Time on A100 | Encoding per frame | Decoding per frame |
> |---|---:|---:|
> | **VTM/H.266 RA mode** | 30.51 s | 19.0 ms |
> | **DCVC-RT** | 5.84 ms | 5.90 ms |
> | **Our VOV** | ~18 min (overfitting) | 2.48 s |
> | **+ 1000-step scaling** | + 27 s (importance sampling) | 25 s |
>
> As our work primarily focuses on framework-level innovation in compression, we do not explicitly optimize runtime in the current version. In addition, functional representations based on per-instance overfitting are inherently more time-consuming than conventional learned codecs. The runtime also scales with spatial resolution and video length, since the Diffusion Transformer requires more denoising steps as the number of visual tokens increases.
>
> As a promising future direction, we could develop an amortized encoder that directly predicts the LoRA adaptation, which could substantially reduce the encoding cost. We will include a discussion in the limitations section on this future direction.
>
> **3. Editability vs compression strength.**
> Thank you for this insightful question. To provide some evidence, we include a qualitative analysis on representative examples. We first evaluate editing with Wan-1.3B across different bitrates
> (https://anonymous.4open.science/r/rebuttal-E61D/edit-wan1.3b-differentbpp.gif).
> As shown, editability does not appear to differ substantially across BPPs. However, the overall editing quality is limited, likely because the model capacity of Wan-1.3B is relatively weak.
>
> We then evaluate a larger video model, Wan-14B
> (https://anonymous.4open.science/r/rebuttal-E61D/edit-wan-14B-differentK.gif), which produces much stronger editing results overall. Due to time constraints, for this model, we analyze the effect of different k values without additional quantization or compression. In the first bear example, all k values give good editing results. In contrast, for the panda example, smaller k leads to worse editing quality, particularly less natural motion.
>
> We also note this issue is difficult to analyze rigorously, since editability can be subjective, and compression strength affects not only editability itself but also the underlying reconstruction quality. As a result, when edit quality degrades, it is difficult to determine whether this is caused by reduced editability or simply by poorer reconstruction. However, overall, these examples suggest that a smaller k  may make editing less robust, although the effect is not completely uniform across examples.
>
> **4. Possible catastrophic mixing.**  We provide additional results in which a single LoRA is fitted to 10 images under two different settings: (1) 10 mutually unrelated images, and (2) 10 images arranged into 5 pairs, where each pair shares a similar prompt or objective. See https://anonymous.4open.science/r/rebuttal-E61D/mixing.png   We do not observe catastrophic mixing in the first setting, despite the larger number of images. In the second setting, however, we do observe mixing some related pairs. This suggests that, in our current setup, the number of images itself is not the main obstacle. Instead, the similarity between prompts/objectives appears to play a more important role in triggering such mixing behavior.
>
> We will include this discussion in the revised version to clarify when catastrophic mixing is more likely to occur.
>
> *We would like to thank you again for the insightful feedback. Your suggestions, particularly on runtime, catastrophic mixing, and editability, have helped us to significantly improve the manuscript in both clarity and completeness. Should you find our response satisfactory, we kindly invite you to consider further raising your rating.*

---

> > ### Author Rebuttal · Reviewer_uXew · 2026-04-02
> >
> > Thanks for authors' detailed reply and experiments. I have no further questions. I will maintain my original score as the encoding and decoding time still has room for improvement.

---

### Official Review · Reviewer_RCu8 · 2026-03-11

**Soundness:** 3
**Presentation:** 4
**Significance:** 3
**Originality:** 3
**Overall Recommendation:** 5
**Confidence:** 4

**Summary:**

The paper introduces "Vision in One Vector" (VOV), a novel framework for extreme low-bitrate visual compression that represents images and videos implicitly as functions rather than explicit arrays of pixels or latent variables, leveraging the rich visual priors of frozen, large-scale diffusion/flow-matching foundation models.

**Compliance With Llm Reviewing Policy:**

Affirmed.

**Final Justification:**

The rebuttal was helpful and increased my confidence in the paper. I maintain my positive score for this paper. I think the proposed method is insightful and promising.

**Key Questions For Authors:**

1. In Appendix B.1, the authors discuss an interesting distortion-perception trade-off achieved via early stopping. Could you clarify what specific stopping time was used for the primary quantitative experiments (e.g., Figure 4)? Are there other possible solutions to further increase the pixel-wise fidelity and avoid hallucination?

2. What is the computational time cost incurred by the inference-time scaling? Intuitively, the importance sampling process across many candidates could be exceedingly slow.

**Limitations:**

yes

**Strengths And Weaknesses:**

**Strengths**:
1. The motivation of leveraging the rich visual priors of diffusion/flow-matching foundation models and ``compressing how to generate the visual signal'' is inspiring and impressive.
2. The paper is well-written and logically structured. The authors provide a clear description of the proposed framework and offer insightful, mathematically grounded discussions connecting their approach to related concepts, such as relative entropy coding and Diff-C.
3. Comprehensive empirical evaluation. The experiments demonstrate competitive perceptual compression performance, provide ablation studies on inference-time scaling and training steps, also includes experiments about bridging compression with generative tasks (e.g., zero-shot editing and merging).

**Weaknesses**:
1. While the conceptual framework is innovative, the proposed pipeline seems to be a combination of several techniques, i.e., LoRA adaptation, hashing trick (Uni-LoRA), learned entropy bottleneck models (Ballé et al., 2018), and inference-time scaling. The authors should more clearly state their novel technical contributions from their system-level integration of these existing tools.
2. No complexity/running time comparison. The paper lacks a comparison of encoding and decoding running times against existing benchmarks. The encoding process is inherently an optimization problem and requires fine-tuning a LoRA module for hundreds of iterations (e.g., 700-1000 steps). The inference-time scaling also introduces extra sampling overhead. Thus, the computational complexity is likely very high.
3. Because the method relies on a stochastic generative prior, it trades pixel-wise fidelity for perceptual realism. As shown in the quantitative results, the PSNR values are noticeably lower than baselines like VTM and DCVC-RT.

---

> ### Author Rebuttal · Authors · 2026-03-30
>
> Thank you for the positive and constructive review. We are glad that you found the overall framework inspiring, the presentation clear, and the empirical study comprehensive. We respond to your concerns below.
>
>
>
> **1. the technical novelty beyond combining existing components.**
> We agree that the paper should state the technical contribution more clearly. We do not claim that LoRA, hashing-style parameter sharing, entropy bottlenecks, or inference-time scaling are individually new. Instead, our contribution is a new compression representation and compression pipeline that integrates these components into a functional representation with strong empirical performance and broader potential. More importantly, we identify and empirically demonstrate a distinctive advantage of this functional representation: unlike conventional compressed representations, it naturally affords greater flexibility at generation time, including the ability to benefit from inference-time scaling.
>
>
> **2. Early stopping used in the main quantitative experiments.**
> Sorry for the confusion. We do **not** use early stopping for the primary quantitative experiments in Figure 4. The early-stopping discussion in Appendix B.1 was intended to develop a more comprehensive understanding on our framework. We will clarify this explicitly in the revision.
>
> 3. > Are there other possible solutions to further increase the pixel-wise fidelity and avoid hallucination?
>
> Yes. We believe this is a promising direction. Actually, one key advantage of the proposed functional representation, as discussed in the conclusion, is its flexibility during the generation (decoding) process.
> In principle, one could further improve pixel-wise fidelity and reduce hallucination by introducing more complicated search strategies during generation, such as beam search or Monte Carlo tree search, to better select trajectories that remain faithful to the target.
> One could also target at better realism by combining the denoising process with external reward or preference models.
>
> These extensions are beyond the main focus of the current paper. We therefore leave a systematic exploration of these directions to future work.
>
>
>
>
> **4. What is the computational time cost incurred by inference-time scaling?**
> Thank you for this helpful suggestion. In practice, the main computational overhead during decoding does not come from the inference-time scaling itself, but from using more denoising steps. This is because the importance-sampling-based scaling does not require additional expensive model evaluations; it mainly involves lightweight Gaussian likelihood computations, which are relatively cheap compared to the denoising network forward passes.
>
> Empirically, we observe the following runtime per frame. This shows the runtime for running the inference with and without scaling with different numbers of steps.
>
> - **No scaling (100 steps):** ~2.48 s / frame
> - **No scaling (1000 steps):** ~25 s / frame
> - **Scaling (1000 steps, 1024 samples per step):** ~27 s / frame
>
> These results show that the additional cost of inference-time scaling is modest. The dominant factor is the number of denoising steps, while the scaling procedure itself adds only a small overhead. We will make this clear in our camera-ready version.
>
> We would like to thank you again for the insightful feedback. We believe our manuscript has been improved a lot following your questions and suggestions.

---

> > ### Author Rebuttal · Reviewer_RCu8 · 2026-04-03
> >
> > The authors have addressed my concerns, especially regarding the inference cost of inference-time scaling. Regarding pixel-wise fidelity, the authors suggested potential ways to improve it via complex generation-time modifications. However, my concern is that the current method, which relies on a stochastic generative prior, may inherently cause fidelity issues and hallucinations (e.g., the PSNR performance is noticeably lower than the baselines).
> > Nevertheless, I will maintain my positive score and lean towards acceptance.

---

> > > ### Author Response · Authors · 2026-04-04
> > >
> > > Thank you for your reply, and thank you for supporting acceptance. We are glad we have addressed most of your concerns. Regarding the fidelity issues and stochastic coding, we want to make a few follow up clarification:
> > >
> > >
> > >
> > >
> > > 1. **The inference time scaling is not complicated and does not involve a large complexity**.
> > >
> > > The main computational cost in our method comes from increasing the number of function evaluations (NFEs), rather than from the scaling algorithm itself. In fact, here we apply scaling in the 100-step setting with 1024 samples
> > >
> > > - No scaling (100 steps): ~2.5 s/frame
> > > - Scaling (100 steps): ~2.8-2.9 s/frame
> > > - No scaling (1000 steps): ~25 s / frame
> > > - Scaling (1000 steps): ~27 s / frame
> > >
> > > Since our scaling approach does not require additional network evaluations, it is quite efficient. We also note that the current implementation uses a simple loop over batches, which could be further optimized.
> > >
> > >
> > > 2. **Our method, combined with lightweight inference-time scaling, already achieves fidelity comparable to GLC-Video, with better realism and visual quality.**
> > >
> > > Its PSNR is lower than VTM, but this is expected due to the Rate-Distortion-Perception(realism) Tradeoff:
> > > codecs that prioritize realism typically sacrifice PSNR. Therefore, it is fairer to compare within codecs within the same regime. Given our substantially better realism and stronger visual results, we believe our method achieves a better overall trade-off than GLC-Video (another codec targeting realism). As a result, we do not think our approach suffers from more severe fidelity degradation or hallucination than similar generative-regime methods such as GLC-Video.
> > >
> > >
> > > As for the approach suggested in our previous reply, we may have misunderstood your question in the earlier rebuttal. The more complicated method we mentioned is intended for cases where one wants to preserve additional properties of the encoded image beyond fidelity alone. We apologize for the confusion.
> > >
> > >
> > > 3. Regarding stochastic coding, we would like to further clarify one extra point:
> > >
> > > **Stochasticity in our method should be viewed as a benefit, not a limitation, especially when realism is important**. As shown in [1], stochastic encoders can achieve better realism than deterministic encoders.
> > >
> > > Also, our work further demonstrates an additional advantage of stochastic coding, namely that **it makes inference-time control / scaling easier**.
> > >
> > >
> > >
> > > [1] Theis, Lucas, and Eirikur Agustsson. "On the advantages of stochastic encoders." arXiv preprint arXiv:2102.09270 (2021).
> > >
> > >
> > >
> > > *We hope our further clarification better addresses your concern.*

---

### Official Review · Reviewer_xG2X · 2026-03-12

**Soundness:** 3
**Presentation:** 3
**Significance:** 3
**Originality:** 3
**Overall Recommendation:** 4
**Confidence:** 3

**Summary:**

This paper introduces a method for representing visual signals, such as images or videos, using a compact vector that parameterizes a LoRA-style adaptation of a pretrained diffusion model. Instead of storing pixels or latent codes, the proposed representation modifies the weights of the diffusion model so that the adapted model reconstructs the visual signal. The paper evaluates the proposed approach through several experiments. Section 4.1 presents an ablation study analyzing the effect of varying the LoRA rank. Section 4.2 reports results on video compression. Section 4.3 explores inference-time scaling strategies, comparing scaling the number of samples versus the number of denoising steps. Section 4.4 demonstrates the possibility of generating tweaked versions of the original image at the decoder.

**Compliance With Llm Reviewing Policy:**

Affirmed.

**Final Justification:**

The rebuttal addressed most of my main concerns and clarified the scope of the paper. I've raised my score to 4.

**Key Questions For Authors:**

* I wonder whether VOV would outperform DDCM and other compression methods leveraging powerful pretrained diffusion models, e.g., for image compression.
* I'm curious to see the quantitative results for experiments in 4.4 and, if possible, comparison against alternative methods. I would encourage the authors to discuss alternatives.

**Limitations:**

Yes

**Strengths And Weaknesses:**

Strengths:
* The perspective of representing data as LoRA updates to a generative model is conceptually interesting.
* The proposed method seems to be effective for extremely low-dimensional representations of visual signals, particularly for videos.
* The experiment setup of generating modified versions of the original image is interesting.
* The paper provides several studies on the variants of the proposed algorithms.

Weaknesses:
* While the perspective of representing data as updates to pretrained diffusion models is interesting, the degree of conceptual or algorithmic novelty appears somewhat limited. For example, LoRA is a commonly used technique. More broadly, leveraging powerful pretrained generative models for image compression has already been explored in prior work such as DDCM [1] (and its recent variant Turbo-DDCM [2]), which is not discussed in the paper.
* Regarding the data compression aspect, although compression is not the only capability of VOV, it is the main setting where quantitative comparisons are provided. In this context, it would be important to compare against compression methods that also leverage pretrained diffusion models, e.g., for images. Otherwise, it is difficult to determine whether the observed benefits come from the strength of the pretrained diffusion model itself or from the proposed VOV representation. Methods such as DDCM would provide informative baselines and help isolate the contribution of the proposed approach.
* Finally, several experimental sections focus primarily on ablation studies (e.g., Sections 4.1 and 4.3) rather than comparisons with alternative methods. Section 4.4 presents qualitative examples only, without comparisons to other approaches that enable image editing or generative variations.

[1] DDCM: Compressed Image Generation with Denoising Diffusion Codebook Models (ICML 2025)
[2] Turbo-DDCM: Fast and Flexible Zero-Shot Diffusion-Based Image Compression (ICLR 2026)

---

> ### Author Rebuttal · Authors · 2026-03-31
>
> Thank you for the thoughtful review. We appreciate that you found the representation perspective interesting and effective. We address your concerns below. Should you find our response satisfactory, we kindly invite you to consider further raising your rating.*
>
> **1. Relation to DDCM / Turbo-DDCM**  Thank you for pointing out these relevant works. We will discuss them more carefully in our final version.
>
> At the same time, we would like to clarify that this is not a fully apples-to-apples comparison:
>
> - DDCM uses a fixed codebook during denoising and encodes the codebook index at each step, whereas our approach aims to learn a continuous functional representation (e.g., which can also be seen from the t-SNE plot for rebuttal to pVCu). This may lead to important downstream differences.
> - VOV compresses the target into a parameter-space adaptation attached to the foundation model. Hence, the encoded representation, together with the base model, is a new diffusion model, which enables more flexible denoising procedures, including inference-time scaling. In contrast, DDCM is closer to a modification of the inference algorithm rather than a parameterized adaptation of the model itself. From this perspective, the two approaches are not necessarily exclusive. In fact, DDCM-style techniques could potentially be incorporated as an inference-time scaling strategy on top of VOV, which we will further demonstrate below.
>
> **2. Comparing DDCM with VOV.**
> We provide an empirical comparison with DDCM on UVG videos in https://anonymous.4open.science/r/rebuttal-E61D/ddcm_vov_avg_metrics_vs_bpp.png. Since our main focus is video compression, a direct comparison with DDCM is not straightforward. To make the comparison as fair as possible, we consider the following settings:
> (1) **DDCM-image**, where each video frame is treated as an individual image and compressed using the official DDCM implementation. Here we use the official repo with SD-2.1-base. Different bpps are obtained by changing the num_noise from 16 tp 128.
>
> (2) **DDCM-video**, where we extend the DDCM compression procedure to the video setting for a fairer comparison. We follow DDCM’s Matching Pursuit strategy. Specifically, we fix num_noise = 1024 , M=12, step = 1000, 2000 for different points. We note that since DDCM was not originally designed for video compression, our extension may still be suboptimal. Nevertheless, we believe this setting provides a more informative and fairer comparison than DDCM-image.
>
> (3) **VOV + DDCM**: As discussed above, DDCM and VOV are not strictly competing approaches and can be combined. We therefore also report results for **VOV + DDCM**. As we use DDCM as a lightweight scaling on top of VOV, we used num_noise=1024 and did not use Matching Pursuit, leading to a negligible increase in bitrate.
>
> As shown, DDCM-image performs substantially worse than VOV, which is expected since it does not capture temporal dependencies across frames.  DDCM-video achieves strong performance, lying between VOV and VOV-scaling. Moreover, VOV + DDCM performs very strongly, with results comparable to VOV-scaling.
>
> Conclusion: empirically, both methods are able to leverage the information in the pretrained model. However, VOV provides a more continuous representation, whereas DDCM is more discrete and inference-algorithm-oriented. As a result, the two approaches are not mutually exclusive and can be combined effectively.
>
> **3. More comparisons beyond ablations.**  Following your suggestion and also the suggestion by reviewer uXew, we have added an INR-based baseline, NVRC, on the UVG dataset: https://anonymous.4open.science/r/rebuttal-E61D/nvrc_vov_metrics_vs_bpp.png.
>
> As shown, VOV achieves stronger performance than NVRC. More importantly, the comparison also highlights our method's visual realism is substantially stronger compared to NVRC. We believe this reflects a key strength of our framework compared to other overfitting-based codecs, as we leverage the information in the pretrained diffusion.
>
> **4. Quantitative results for editing experiments in 4.4.**
>
> The experiments in Sec. 4.4 are intended as demonstrative evaluations of the editing ability of our functional representation. Importantly, our method enables editing with based text-guided diffusion models that originally do not accept visual inputs. To demonstrate this ability more comprehensively, we present a compelling video editting and merging result with Wan-14B at https://anonymous.4open.science/r/rebuttal-E61D/edit-video.gif. While these results are primarily qualitative, they highlight a novel functionality enabled by our method. Despite not the main focus of our paper, we emphasize that such editing capabilities are complementary to our primary objective of compression.
>
>
> *We would like to thank you again for the insightful feedback. Your suggestions, particularly on the related works, have helped us to significantly improve the manuscript's completeness.*

---

> > ### Author Rebuttal · Reviewer_xG2X · 2026-04-06
> >
> > I would like to thank the authors for the additional experimental results and for clarifying some of the differences between VOV and DDCM, e.g. discrete vs. continuous representations. The work of DDCM+VOV seems interesting. Some of my concerns still remain though: there are experiments of VOV on images in the paper, and I do not fully understand why VOV and DDCM cannot be compared in the image experiments. Experiments on videos are, as the authors note, "As shown, DDCM-image performs substantially worse than VOV, which is expected since it does not capture temporal dependencies across frames."

---

> > > ### Author Response · Authors · 2026-04-06
> > >
> > > Thanks for your reply.
> > >
> > > We would like to further clarify the scope of our work.
> > >
> > >
> > > Our paper proposes methods for both *representation* and *compression*. The image experiments in our manuscript are intended to demonstrate the representation part, rather than to make image compression claims.
> > >  **On the compression side, our focus is specifically on video, where we achieve state-of-the-art performance on perceptual video compression.**
> > >
> > > Since image and video compression differ substantially in scale and temporal structure, although the overall presentation framework applies to both modalities, **image compression is not within the scope or claim of our paper**.
> > >
> > >
> > > Therefore, we put more effort into implementing DDCM (an image codec) to video modality for comparison.
> > > We considered two video-based evaluation settings for DDCM. The first is a direct frame-wise application of DDCM to videos ("DDCM-image"), where each frame is encoded independently. The second, "DDCM-video", is our extension of the DDCM procedure to the video setting, which we believe is a fairer comparison for evaluating video compression performance. Under both settings, our approach outperforms the baseline.
> > >
> > > We also emphasize that our rebuttal discussion of temporal dependencies specifically concerns the frame-wise "DDCM-image" setting, while DDCM-video already provides a more appropriate and fair video-level baseline.
> > >
> > > We hope this clarification helps place our paper in a fairer context for evaluation.

---

### Official Review · Reviewer_pVCu · 2026-03-16

**Soundness:** 4
**Presentation:** 3
**Significance:** 3
**Originality:** 3
**Overall Recommendation:** 4
**Confidence:** 3

**Summary:**

This paper proposes an implicit image/video compression method built on top of large pretrained generative models: Qwen for images and Wan for videos.

The main idea behind “Vision in One Vector” is to compress a specific image or video into a single vector with on the order of 100,000 quantized scalars.

This vector is meant to represent the entire LoRA adaptation of the model, i.e. effectively all the A and B matrices across all adapted layers.

The way they do this is by reconstructing all of these LoRA parameters from one shared vector through a fixed random mapping generated from a PRNG, so they do not need to explicitly store the full set of LoRA parameters. They then optimize this vector on the target image or video, using a caption obtained from a separate captioning model, under the standard generative training objective of the pretrained model. To further reduce bitrate, the vector is quantized to a small number of bits per entry.

The paper also proposes an inference-time scaling method based on importance sampling. At each denoising step, it draws multiple candidate samples, score them according to how well they match the target relative to the model proposal, and keep one branch. This extra branch-and-select procedure improves performance further.

This “vision vector” then becomes the representation of the target content in its own right. The paper shows video compression experiments where the method looks strong on perceptual metrics, and in those metrics it outperforms H.265 and H.266 baselines in the reported setting, although it is still not competitive on PSNR.

Finally, the paper shows that the adapted model can still do editing: the vector seems to capture the identity/content of the target image or video, while the text prompt can still change aspects of the scene. This only works as long as the model is not overfit too hard to the target.

**Compliance With Llm Reviewing Policy:**

Affirmed.

**Final Justification:**

I have read the rebuttals which have cleared all my concerns. I recommend acceptance for this paper. Although, my limited familiarity with the field of compression hampers my ability to judge it's significance fully.

**Key Questions For Authors:**

1. **Provide the full conventional-codec configuration used for the HM/VTM baselines. And, comparing agianst the best settings of the convention compression baselines.** A clear answer here would increase my confidence in the fairness of the comparison.
2. **Quantify the tradeoff between optimization length, reconstruction quality, and editability more explicitly.** This is not as important but it will make the paper more complete.

**Limitations:**

yes

**Strengths And Weaknesses:**

Note about me: I'm not entirely in the compression community, I'm more of a diffusion person who cares about compression. I'm not fully qualified to judge the originality of the work.

**Strengths**

- As someone who at some point tried to work on a related video compression idea and then stopped, I think the paper pushes this direction much farther than I would have expected.
- The core idea is genuinely clever: represent the full LoRA adaptation across all tuned layers using a single shared vector together with a fixed PRNG-generated projection.
- I like that the authors were very careful on hitting all the possible compression bottlenecks:
    - first the one-vector adaptation,
    - then quantization-aware optimization,
    - then additional inference-time scaling through branch-and-select importance sampling
- I found the scaling result particularly interesting. Figure 6 shows that without branching, increasing the number of denoising steps gives almost no improvement, while multi-sample branching improves performance substantially. This is a genuinely interesting result.
    - Why? Because our community often treats deterministic / ODE-style sampling as the obvious default for content generation and for these models. This implicit bias makes it hard to realize the gain “only” possible from the SDE formulation.
- I also appreciated the distortion-perception discussion. I think the paper does a good job of implicitly showing the limitation of PSNR for this kind of generative/perceptual compression setting which help the community steers away from optimizing to the wrong and pathological-for-generative-approaches kind of metric.

**Weaknesses**

- My main concern is the baseline comparison to conventional codecs.
    - Including H.266/VTM is the right choice, and VTM-RA is certainly a strong baseline, but the paper does not give enough detail about the exact codec configuration.
    - For reproducibility and fairness (because different compression presets matter a lot), I would want the full settings spelled out more clearly.
    - In particular, the paper uses HM-RA / VTM-RA, but does not explain why random-access mode was chosen, or how this compares to stronger or less constrained practical settings.
- To be clear, based on the qualitative videos, I am inclined to believe the proposed method really is producing unusually sharp perceptual results. Still, I would like to see a cleaner and more transparent conventional-codec comparison.
- I would also like to see a more quantitative study of optimization length. The paper discusses the tradeoff between fitting the target and preserving editability, which is important, but from a pure compression point of view it would still be useful to know:
    - how much is gained by optimizing longer,
    - where saturation happens,
    - and how editability degrades as a function of optimization time.
- A more forward-looking concern is that the title “Vision in One Vector” suggests something broader than compression and editing alone. “Vision” also suggests some amount of understanding. For example:
    - can the vector support classification,
    - can one do linear probing on it,
    - does it capture anything beyond target-specific reconstruction/editing information?
- My guess is that much of the vector is low-level and reconstruction-oriented, in which case the title may be somewhat broader than what is actually demonstrated.

**Clarity / presentation issues**

- The quantization part is underexplained.
    - The paper says the one-vector adaptation can be quantized to approximately 1–3 bits per parameter, but it is not clear what exact operating points are used in practice, nor how much performance is lost relative to the non-quantized version.
    - This part deserves a clearer quantitative discussion.
- It took me some effort to locate the exact base models used in the experiments. This should be easier to find in the main text.
    - The exact models are mostly specified in Appendix E rather than clearly surfaced upfront: Qwen-Image-20B for images and Wan-2.1-1.3B for videos.
- The notation such as ($1000 \times 2^{10}$) is not self-explanatory. It should be explained explicitly in the caption as “number of denoising steps × number of samples per step.”
- More generally, the distinction between number of denoising steps and number of samples per step is not immediately clear on first reading. More distinct terminology would help.
- The paper also mentions settings like ($2^{18}$) samples per step rather casually. This is an extremely large number, and I think the paper should be more explicit about whether this is meant as a practical setting, a stress test, or simply an upper-bound scaling experiment.

---

> ### Author Rebuttal · Authors · 2026-03-30
>
> Thank you for the detailed and constructive review. We are especially encouraged that you found our work genuinely clever, interesting, and meaningful. We respond to your concerns below.
>
> **1. Traditional codec configurations.**  We used the reference configuration files recommended by standardization organizations. Specifically, for VTM, the cfg file is from https://vcgit.hhi.fraunhofer.de/jvet/VVCSoftware_VTM/-/blob/master/cfg/encoder_randomaccess_vtm.cfg?ref_type=heads ; and for HM, we use cfg file from https://vcgit.hhi.fraunhofer.de/jvet/HM/-/blob/master/cfg/encoder_randomaccess_main10.cfg?ref_type=heads .The used random access mode help to produce the strongest performance from H.266/VTM or H.265/HM, which is a common setting in papers on neural video compression.  We will include the full configuration details in the camera-ready version.
>
>
> **2. Optimization length vs. reconstruction.**
> Thank you for this insightful suggestion. We provide PSNR-iteration curves for the compression at https://anonymous.4open.science/r/rebuttal-E61D/iter_compression.png. Note that in compression experiments, we first warm-up the adaptation and then impose the bitrate constraint. The curves correspond to this second stage.
>
> We report results for two bitrates. The plots show an interesting pattern. For VOV without scaling, longer optimization helps the method satisfy the rate constraint more accurately, and the performance begins to plateau after roughly 750–1000 iterations. In contrast, with VOV-scaling, the reconstruction quality continues to improve even beyond 750 iterations.
>
> Our interpretation is that longer optimization makes the proposal closer to the target distribution in our scaling setting. But a better fit in this sense does not always directly translate to better generation quality (similar to diffusion, a better ELBO does not always mean better quality in generation). This is why longer optimization appears to be especially helpful for the scaling setting rather than for plain VOV.
>
>
> **3. Optimization length vs. editability.**  We agree that this is another interesting direction. However, we would like to first clarify a potential misunderstanding. What we observe is not exactly a decrease in editability, but rather a decrease in general generation ability.
>
> Specifically, as the model becomes increasingly overfitted to the target image, samples generated from partially related prompts become more similar to that overfitted image. We provide two examples in https://anonymous.4open.science/r/rebuttal-E61D/overfit_generation_ability.png.
> For the editing experiments, this is less problematic because the goal is precisely to generate images related to the target. However, for a more general generation, one may prefer broader diversity, and this is where the trade-off appears. We will ensure to clarify this better in our final version.
>
> **4. Scope of the title**
> We agree that the current title may be broader than what is directly demonstrated in the paper. We will make this scope clearer and adjust the title accordingly in the final version. Please let us know if you have more suggestions regarding the title.
>
> Regarding whether the learned vector supports downstream tasks, we currently do not have sufficient training or evaluation data to make a strong claim. As a preliminary analysis, we projected the vectors obtained from 19 video clips using t-SNE. As shown in https://anonymous.4open.science/r/rebuttal-E61D/tsne.png, the embeddings do exhibit some structure: videos with similar style or objective tend to cluster more closely than unrelated ones. This suggests that the representation captures some meaningful information, although we agree that it is more consistent with relatively low-level structure than with broad semantic understanding.
>
> **5. Quantization clarification.**   Following [1], during training, we add uniform noise to mimic quantization error. As a result, we do not observe a visible quality drop after actual quantization. We will make this point explicit in the revised version.
>
> [1] Ballé, Johannes, Valero Laparra, and Eero P. Simoncelli. "End-to-end optimized image compression." ICLR 2017.
>
> **6. Large sample counts per step.**   The very large sample count is intended primarily as a stress test to study the scaling behavior of the method, rather than as the main practical operating point. In practice, a much smaller number such as 1024 samples is more realistic. We will clarify this distinction in the paper.
>
> **7. Clarity/presentation issues.**
> We agree with your comments on clarity and presentation, and we will revise the manuscript accordingly to make these points more explicit and easier to follow.
>
> *We would like to thank you again for the insightful feedback. Your suggestions have helped us to significantly improve the manuscript in both clarity and completeness. Should you find our response satisfactory, we kindly invite you to consider further raising your rating.*

---

> > ### Author Rebuttal · Reviewer_pVCu · 2026-04-04
> >
> > 1. Thanks for including the specs. I agree that this setting is common. However, I still do not feel confident that it is the best setting for those codecs, and I believe that running results against the best settings would only strengthen the paper’s contribution while also not taking much more time.
> > 2. This is interesting. I particularly like the fact that the scaling trick compounds with longer training.
> > 4. I still think the title should be changed.
> >
> > I still recommend that this paper be accepted.

---

> > > ### Author Response · Authors · 2026-04-04
> > >
> > > Thank you for the follow-up. We clarify our points below.
> > >
> > > 1. We clarify that both VTM and HM are used with the official Random Access (RA) configurations released by the standardization committees. These are widely regarded as the strongest reference settings for rate–distortion performance in academic evaluations, due to their hierarchical GOP structure and full bidirectional prediction. For example, the HM reference implementation is different and significantly stronger than practical encoders such as x265 that is optimized for compression efficiency rather than speed. Therefore, **our comparison is already against best-practice and upper-bound conventional codec performance**. We will make this clarification explicit in the final version.
> > >
> > > 2. We are glad you found this result interesting. We will discuss this observation more clearly in the final version. Thank you for suggesting this additional analysis.
> > >
> > > 3. Yes, we agree that the title should be changed, and we will revise it in the final version. At the moment, we are considering options such as "Compression in One Vector", "Compression as Adaptation", or simply "Implicit Visual Compression with Diffusion Foundation Models". We appreciate your suggestion here, as we agree the revised title should better match the actual scope of the paper.
> > >
> > > Thank you again for your insightful reply. We hope this response clarifies our points more clearly and helps address your concerns.

---

### Decision · Program_Chairs · 2026-04-30

**Decision:**

Accept (regular)

**Comment:**

The paper introduces an fascinating idea: compressing visual information into a single-vector. To achieve that, the paper leverages low-rank adaptation of a frozen generative model to implicitly represent images and videos.  This enables very strong compression of the signal. Originally, the reviewer raised concerns, e.g. over novelty (xG2X) or baseline comparisons (pVCu), which were later addressed by the rebuttal/response. After the discussion period, the reviewers believe this paper is ready to be accepted to the conference, conditioned on the authors addressing the concerns in the camera-ready version.

Please check manually the references in the camera-ready version. There are hallucinated references, including a paper title that I could not even be found manually.